# Sparsifying Bayesian neural networks with latent binary variables and normalizing flows

## Abstract

Artificial neural networks are powerful machine learning methods used in many modern applications. A common issue is that they have millions or billions of parameters, and therefore tend to overfit. Bayesian neural networks (BNN) can improve on this since they incorporate parameter uncertainty. Latent binary Bayesian neural networks (LBBNN) further take into account structural uncertainty by allowing the weights to be turned on or off, enabling inference in the joint space of weights and structures. Mean-field variational inference is typically used for computation within such models. In this paper, we will consider two extensions of variational inference for the LBBNN: Firstly, by using the local reparametrization trick (LCRT), we improve computational efficiency. Secondly, and more importantly, by using normalizing flows on the variational posterior distribution of the LBBNN parameters, we learn a more flexible variational posterior than the mean field Gaussian. Experimental results on real data show that this improves predictive power compared to using mean field variational inference on the LBBNN method, while also obtaining sparser networks. We also perform two simulation studies. In the first, we consider variable selection in a logistic regression setting, where the more flexible variational distribution improves results. In the second study, we compare predictive uncertainty based on data generated from two-dimensional Gaussian distributions. Here, we argue that our Bayesian methods lead to more realistic estimates of predictive uncertainty.

## 1 Introduction

Modern deep learning architectures can have billions of trainable parameters (Khan et al., 2020). Due to the large number of parameters in the model, the network can overfit, and therefore may not generalize well to unseen data. Further, the large number of parameters gives computational challenges both concerning training the network and for prediction. The lottery ticket hypothesis (Frankle & Carbin, 2018; Pensia et al., 2020) states that dense networks contain sparse subnetworks that can achieve the performance of the full network. Even though the construction/training of such subnetworks is not necessarily simpler than training the full network, substantial gains can be obtained in the prediction stage when using sparse subnetworks.

Bayesian neural networks (BNN, Neal, 1992; MacKay, 1995; Bishop, 1997) use a rigorous Bayesian methodology to handle parameter and prediction uncertainty. In principle, prior knowledge can be incorporated through appropriate prior distributions, but most approaches within BNN so far only apply very simple convenience priors (Fortuin, 2022). However, approaches where knowledge-based priors are incorporated are starting to appear (Tran et al., 2022; Sam et al., 2024). Due to a more proper procedure for handling uncertainty, the Bayesian approach does, in many cases, result in more reliable solutions with less overfitting and better uncertainty measures. However, this comes at the expense of extremely high computational costs. Until recently, inference on Bayesian neural networks could not scale to large multivariate data due to limitations of standard Markov chain Monte Carlo (MCMC) approaches, the main quantitative procedure used for complex Bayesian inference. Recent developments of variational Bayesian approaches (Gal, 2016) allow us to approximate the posterior of interest and lead to more scalable methods.

In this work, we consider a formal Bayesian approach for obtaining sparse subnetworks by including latent binary variables allowing the weights in the network to be turned on or off. This allows us to model structural uncertainty and makes BNNs more robust to misspecification (Papamarkou et al., 2024). This also opens the door for Bayesian model selection and averaging (Hubin & Storvik, 2019). The computational procedure is based on variational inference. Earlier work (Hubin & Storvik, 2019; Bai et al., 2020; Hubin & Storvik, 2024), have considered similar settings approaches, our main contributions are

- improvements of computational efficiency in LBBNNs through the use of the local parametrization trick (Kingma et al., 2015);

- extending the class of variational distributions to normalizing flows allowing modeling dependencies;

- demonstrating improvements in predictive power, sparsity, and variable selection through experiments on real and simulated data;

- demonstrating robust performance in uncertainty quantification through the expected calibration error for classification and Pinball loss for regression.

## 1.1 Literature background

The idea of using a mathematical model to imitate how the brain works was first introduced in McCulloch & Pitts (1943). However, it was not until more recent years that the true power of these models could be harnessed with the idea of using backpropagation (Rumelhart et al., 1986) to train the model with gradient descent. With the advent of modern GPU architectures, deep neural networks can be scaled to big data, and have shown to be very successful on a variety of tasks including computer vision (Voulodimos et al., 2018), and natural language processing (Young et al., 2018). Modern deep learning architectures can have billions of trainable parameters (Khan et al., 2020). Due to the large number of parameters in the model, the network has the capacity to overfit, and therefore may not generalize well to unseen data. Various regularization methods are used to try to deal with this, such as early stopping (Prechelt, 1998), dropout (Srivastava et al., 2014) or data augmentation (Shorten & Khoshgoftaar, 2019). These techniques are heuristic and therefore it is not always clear how to use them and how well they work in practice. It is also possible to reduce the number of parameters in the network with pruning. This is typically done with the dense-to-sparse method (Han et al., 2017). Here, a dense network is trained, while the importance of the weights (i.e. their magnitude) is recorded. Then, the weights that fall below the sparsity threshold (a hyperparameter) are removed. In Frankle & Carbin (2018), it is hypothesized that in randomly initialized dense networks, there exists a sparse sub-network (the winning lottery ticket) that can obtain the same test accuracy as the original dense network. Instead of training and pruning once, referred to as one-shot pruning, this process is repeated sequentially several times, removing a certain percentage of the remaining weights each time, which then results in networks that have a higher degree of sparsity than the ones found with one-shot pruning. However, this comes at a higher computational cost. Further refinements to this are done in Evci et al. (2020), where the network starts off dense, and dynamically removes the weights with the smallest magnitude, while also adding new connections based on gradient information. Again, these approaches are heuristic and lack a solid theoretical foundation. Another issue with deep learning models is that they often make overconfident predictions. In Szegedy et al. (2013), it was shown that adding a small amount of noise to an image can trick a classifier into making a completely wrong prediction (with high confidence), even though the image looks the same to the human eye. The opposite is also possible, images that are white noise can be classified with almost complete certainty to belong to a specific class (Nguyen et al., 2015).

Bayesian neural networks (BNNs) were presented by Neal (1992), MacKay (1995), and Bishop (1997). They use a rigorous Bayesian methodology to handle parameter and prediction uncertainty. In many cases, this results in more reliable solutions with less overfitting. Still, BNNs tend to be heavily over-parameterized and difficult to interpret. It is therefore interesting to consider sparsity-inducing methods from a Bayesian perspective. This is typically done by using sparsity-inducing priors, as in variational dropout (Kingma et al., 2015; Molchanov et al., 2017), which uses the independent log uniform prior on the weights. This is an improper prior, meaning that it is not integrable and thus not a valid probability distribution. As noted in Hron et al. (2017), using this prior, combined with commonly used likelihood functions leads to an improper

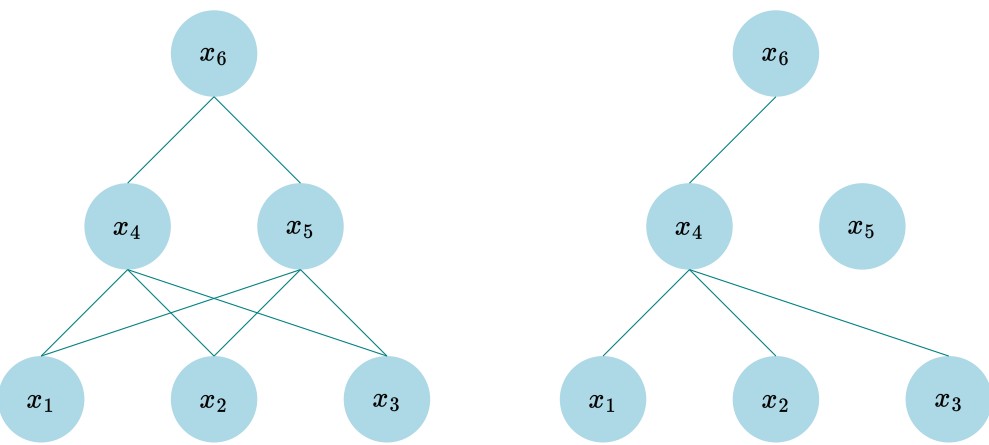

Figure 1: A dense network on the left, one possible sparse structure on the right.

posterior, meaning that the obtained results can not be explained from a Bayesian modeling perspective. It is argued that variational dropout should instead be interpreted as penalized maximum likelihood estimation of the variational parameters. Additionally, Gale et al. (2019) finds that while variational dropout works well on smaller networks, it gets outperformed by the heuristic (non-Bayesian) methods on bigger networks. Another type of sparsity inducing prior is the independent scale mixture prior, where Blundell et al. (2015) proposed a mixture of two Gaussian densities, where using a small variance for the second mixture component leads to many of the weights having a prior around 0. Another possibility is to use the independent spike-and-slab prior, most commonly used in Bayesian linear regression models. This prior is used in latent binary Bayesian neural networks (LBBNN) introduced by Hubin & Storvik (2019; 2024) and concurrently in Bai et al. (2020). The spike-and-slab prior for a special case of LBBNN with the ReLu activation function was studied from a theoretical perspective in Polson & Ročková (2018). In Hubin & Storvik (2019) it was empirically shown that using this prior will induce a very sparse network (around 90 % of the weights were removed) while maintaining good predictive power. Using this approach thus takes into account uncertainty around whether each weight is included or not (structural uncertainty) and uncertainty in the included weights (parameter uncertainty) given a structure, allowing for a fully (variational) Bayesian approach to network sparsification (see Figure 1). In this paper, we show that transforming the variational posterior distribution with normalizing flows can result in even sparser networks while improving predictive power compared to the mean field approach used in Hubin & Storvik (2019). Additionally, we demonstrate that the flow network handles predictive uncertainty well, and performs better than the mean-field methods at variable selection in a logistic regression setting with highly correlated variables, thus demonstrating higher quality in structure learning.

## 2 The model

Given the explanatory variable $\boldsymbol{x} \in \mathbb{R}^n$, and the response variable $\boldsymbol{y} \in \mathbb{R}^m$, a neural network models the function

$$\boldsymbol{y} \sim f(\cdot; \boldsymbol{\eta}(\boldsymbol{x}))$$

where the distribution $f(\cdot; \boldsymbol{\eta})$ is parameterised by the vector $\boldsymbol{\eta}$. The vector $\boldsymbol{\eta}$ is obtained through a composition of semi-affine transformations:

$$u_j^{(l)} = \sigma^{(l)} \left( \sum_{i=1}^{n^{(l-1)}} u_i^{(l-1)} \gamma_{ij}^{(l)} w_{ij}^{(l)} + b_j^{(l)} \right), j = 1, \ldots, n^{(l)}, l = 1, \ldots, L, \tag{1}$$

with $\eta_j = u_j^{(L)}$. Additionally, $\boldsymbol{u}^{(l-1)}$ denotes the inputs from the previous layer (with $\boldsymbol{u}^0 = \boldsymbol{x}$ corresponding to the explanatory variables), the $w_{ij}^{(l)}$'s are the weights, the $b_j^{(l)}$'s are the bias terms, and $n^{(l)}$ (and $n^{(0)} = n$)

the number of inputs at layer $l$ of a total $L$ layers. Further, we have the elementwise non-linear activation functions $\sigma^{(l)}$. The additional parameters $\gamma_{ij}^{(l)} \in \{0, 1\}$ denote binary inclusion variables for the corresponding weights.

Following Polson & Ročková (2018); Hubin & Storvik (2019); Bai et al. (2020), we consider a *structure* to be defined by the configuration of the binary vector $\boldsymbol{\gamma}$, and the weights of each structure conditional on this configuration. To consider uncertainty in both structures and weights, we use the spike-and-slab prior, where for each (independent) layer $l$ of the network, we also consider the weights to be independent:

$$p(w_{ij}^{(l)}|\gamma_{ij}^{(l)}) = \gamma_{ij}^{(l)}\mathcal{N}(w_{ij}^{(l)}; 0, (\sigma^{(l)})^2) + (1 - \gamma_{ij}^{(l)})\delta(w_{ij}^{(l)})$$
$$p(\gamma_{ij}^{(l)}) = \text{Bernoulli}(\gamma_{ij}^{(l)}; \alpha^{(l)}).$$

We will use the nomenclature from Hubin & Storvik (2019) and refer to this as the LBBNN model. Here, $\delta(\cdot)$ is the Dirac delta function, which is considered to be zero everywhere except for a spike at zero. In addition, $\sigma^2$ and $\alpha$ denote the prior variance and the prior inclusion probability of the weights, respectively. In practice, we use the same variance and inclusion probability across all the layers and weights, but this is not strictly necessary. In principle, one can incorporate knowledge about the importance of individual covariates or their co-inclusion patterns by adjusting the prior inclusion probabilities for the input layer or specifying hyper-priors. This is common in Bayesian model selection literature (Fletcher & Fletcher, 2018), but not yet within BNNs.

## 3 Bayesian inference

The main motivation behind using LBBNNs is that we are able to take into account both structural and parameter uncertainty, whereas standard BNNs are only concerned with parameter uncertainty. By doing inference through the posterior predictive distribution, we average over all possible structural configurations, and parameters. For a new observation $\tilde{\boldsymbol{y}}$ given training data, $\mathcal{D}$, we have:

$$p(\tilde{\boldsymbol{y}}|\mathcal{D}) = \sum_{\boldsymbol{\gamma}} \int_{\boldsymbol{w}} p(\tilde{\boldsymbol{y}}|\boldsymbol{w}, \boldsymbol{\gamma}, \mathcal{D})p(\boldsymbol{w}, \boldsymbol{\gamma}|\mathcal{D}) \, d\boldsymbol{w}.$$

This expression is intractable due to the ultra-high dimensionality of $\boldsymbol{w}$ and $\boldsymbol{\gamma}$, and using Monte Carlo sampling as an approximation is also challenging due to the difficulty of obtaining samples from the posterior distribution, $p(\boldsymbol{w}, \boldsymbol{\gamma}|\mathcal{D})$. Instead of trying to sample from the true posterior, we turn it into an optimization problem, using variational inference (VI, Blei et al., 2017). The key idea is that we replace the true posterior distribution with an approximation, $q_{\boldsymbol{\theta}}(\boldsymbol{w}, \boldsymbol{\gamma})$, with $\boldsymbol{\theta}$ denoting some variational parameters. We learn the variational parameters that make the approximate posterior as close as possible to the true posterior. Closeness is measured through the Kullback-Leibler (KL) divergence,

$$\text{KL}\left[q_{\boldsymbol{\theta}}(\boldsymbol{w}, \boldsymbol{\gamma})||p(\boldsymbol{w}, \boldsymbol{\gamma}|\mathcal{D})\right] = \sum_{\boldsymbol{\gamma}} \int_{\boldsymbol{w}} q_{\boldsymbol{\theta}}(\boldsymbol{w}, \boldsymbol{\gamma}) \log \frac{q_{\boldsymbol{\theta}}(\boldsymbol{w}, \boldsymbol{\gamma})}{p(\boldsymbol{w}, \boldsymbol{\gamma}|\mathcal{D})} \, d\boldsymbol{w}. \tag{2}$$

Minimizing the KL-divergence (with respect to $\boldsymbol{\theta}$) is equivalent to maximizing the evidence lower bound (ELBO):

$$\text{ELBO}(q_{\boldsymbol{\theta}}) = \mathbb{E}_{q_{\boldsymbol{\theta}}(\boldsymbol{w}, \boldsymbol{\gamma})}\left[\log p(\mathcal{D}|\boldsymbol{w}, \boldsymbol{\gamma})\right] - \text{KL}\left[q_{\boldsymbol{\theta}}(\boldsymbol{w}, \boldsymbol{\gamma})||p(\boldsymbol{w}, \boldsymbol{\gamma})\right]. \tag{3}$$

The objective is thus to maximize the expected log-likelihood while penalizing with respect to the KL divergence between the prior and the variational posterior. How good the approximation becomes depends on the family of variational distributions $\{q_{\boldsymbol{\theta}}, \boldsymbol{\theta} \in \Theta\}$ that is chosen.

### 3.1 Choices of variational families

A common choice (Blundell et al., 2015) for the approximate posterior in (dense) Bayesian neural networks is the mean-field Gaussian distribution. For simplicity of notation, denote now by $\mathbf{W}$ the set of weights corresponding to a specific layer. Note that from here on, we drop the layer notation for readability, since

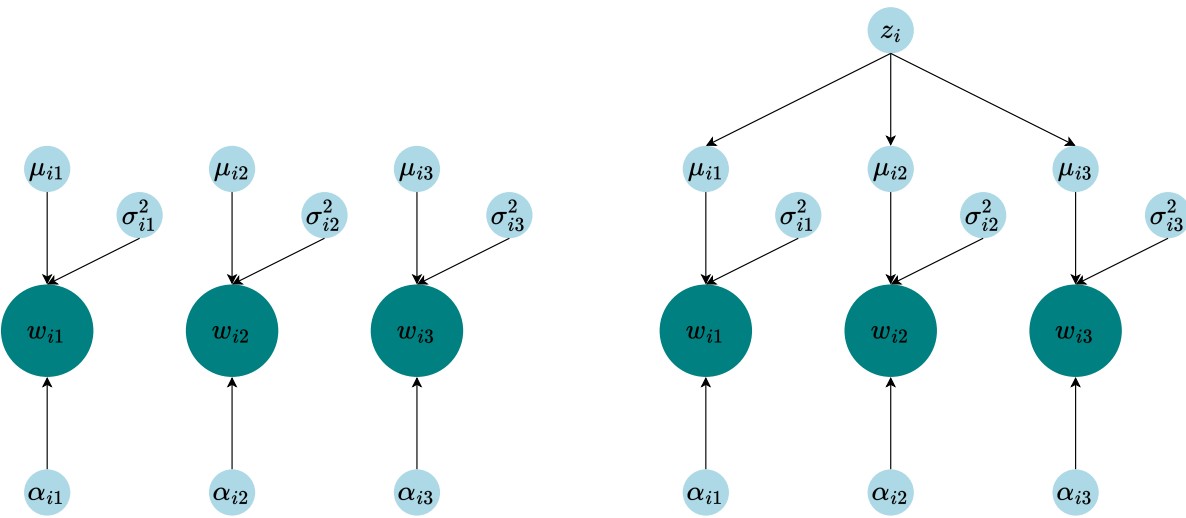

Figure 2: On the left, the mean-field variational posterior where the weights are assumed independent. On the right, the latent variational distribution $z$ allows for modeling dependencies between the weights.

the parameters at different layers will always be considered independent in both the variational distribution and the prior. Then

$$q_{\boldsymbol{\theta}}(\mathbf{W}) = \prod_{i=1}^{n_{in}} \prod_{j=1}^{n_{out}} \mathcal{N}(w_{ij}; \tilde{\mu}_{ij}, \tilde{\sigma}_{ij}^2),$$

where $n_{in}$ and $n_{out}$ denote the number of neurons in the previous and current layer, respectively. Weights corresponding to different layers are assumed independent as well. The mean-field Gaussian distribution for Bayesian neural networks can be extended to include the binary inclusion variables following Carbonetto & Stephens (2012):

$$q_{\boldsymbol{\theta}}(\mathbf{W}|\boldsymbol{\Gamma}) = \prod_{i=1}^{n_{in}} \prod_{j=1}^{n_{out}} \left( \gamma_{ij} \mathcal{N}(w_{ij}; \tilde{\mu}_{ij}, \tilde{\sigma}_{ij}^2) + (1 - \gamma_{ij}) \delta(w_{ij}) \right);$$

$$q_{\tilde{\alpha}_{ij}}(\gamma_{ij}) = \text{Bernoulli}(\gamma_{ij}; \tilde{\alpha}_{ij}).$$

(4)

Here, $\boldsymbol{\Gamma}$ is the set of inclusion indicators corresponding to a specific layer. However, the mean-field Gaussian distribution (Blundell et al., 2015) is typically too simple to be able to capture the complexity of the true posterior distribution. We follow Ranganath et al. (2016), and introduce a set of auxiliary latent variables $\boldsymbol{z}$ to model dependencies between the weights in $q$, and use the following variational posterior distribution:

$$q_{\boldsymbol{\theta}}(\mathbf{W}|\boldsymbol{\Gamma}, \boldsymbol{z}) = \prod_{i=1}^{n_{in}} \prod_{j=1}^{n_{out}} \left( \gamma_{ij} \mathcal{N}(w_{ij}; z_i \tilde{\mu}_{ij}, \tilde{\sigma}_{ij}^2) + (1 - \gamma_{ij}) \delta(w_{ij}) \right);$$

$$q_{\tilde{\alpha}_{ij}}(\gamma_{ij}) = \text{Bernoulli}(\gamma_{ij}; \tilde{\alpha}_{ij}),$$

(5)

where $\boldsymbol{z} = (z_1, ..., z_{n_{in}})$ follows a distribution $q_{\boldsymbol{\phi}}(\boldsymbol{z})$. For an illustration of the difference between the two variational distributions in Equation (4) and Equation (5), see Figure 2. The novelty in our suggested variational distribution is to combine both weight and structural uncertainty, in addition to modeling dependencies between the weights. As for $\mathbf{W}$, also $\boldsymbol{z}$ is a set of variables related to a specific layer and independence between layers is assumed also for $\boldsymbol{z}$'s. To increase the flexibility of the variational posterior, we apply normalizing flows (Rezende & Mohamed, 2015) to $q_{\boldsymbol{\phi}}(\boldsymbol{z})$. In general, a normalizing flow is a composition of invertible transformations of some initial (simple) random variable $\mathbf{z}_0$,

$$\boldsymbol{z}_k = f_k(\boldsymbol{z}_{k-1}), \quad k = 1, ..., K.$$

The log density of the transformed variable $\boldsymbol{z} = \boldsymbol{z}_K$ is given as,

$$\log q(\boldsymbol{z}) = \log q_0(\boldsymbol{z}_0) - \sum_{k=1}^{K} \log \left| \det \frac{\partial \boldsymbol{z}_k}{\partial \boldsymbol{z}_{k-1}} \right|. \tag{6}$$

We are typically interested in transformations that have a Jacobian determinant that is tractable, and fast to compute, in addition to being highly flexible. Transforming the variational posterior distribution in a BNN with normalizing flows was first done in Louizos & Welling (2017), who coined the term multiplicative normalizing flows (BNN-FLOW), where the transformations were applied in the activation space instead of the weight space. As the weights are of much higher dimensions, the number of flow parameters and thus the number of parameters of variational distribution would explode quickly. We will follow Louizos & Welling (2017) here. The main difference in our work is that by using the variational posterior in Equation (5), we also get sparse networks.

For the normalizing flows, we will use the inverse autoregressive flow (IAF), with numerically stable updates, introduced by Kingma et al. (2016). It works by transforming the input in the following way:

$$\begin{aligned}
\boldsymbol{z}_{k-1} &= \text{input} \\
\boldsymbol{m}_k, \boldsymbol{s}_k &= g(\boldsymbol{z}_{k-1}) \\
\boldsymbol{\kappa}_k &= \text{sigmoid}(\boldsymbol{s}_k) \\
\boldsymbol{z}_k &= \boldsymbol{\kappa}_k \odot \boldsymbol{z}_{k-1} + (1 - \boldsymbol{\kappa}_k) \odot \boldsymbol{m}_k,
\end{aligned} \tag{7}$$

where $g$ is a neural network and $\odot$ denotes elementwise multiplication. Assuming the neural network in Equation (7) is autoregressive (i.e $z_{k,i}$ can only depend on $z_{k,1:i-1}$), we get a lower triangular Jacobian and

$$\log \left| \det \frac{\partial \boldsymbol{z}_k}{\partial \boldsymbol{z}_{k-1}} \right| = \sum_{i=1}^{n_{in}} \log \kappa_{k,i}. \tag{8}$$

## 3.2 Computing the variational bounds

Minimization of the KL in Equation (2) is difficult due to the introduction of the auxiliary variable $\boldsymbol{z}$ in the variational distribution. In principle, $\boldsymbol{z}$ could be integrated out, but in practice this is difficult. Following Ranganath et al. (2016), we instead introduce $\boldsymbol{z}$ as an auxiliary variable also in the posterior distribution by defining

$$p(\boldsymbol{w}, \boldsymbol{\gamma}, \boldsymbol{z} | \mathcal{D}) = p(\boldsymbol{w}, \boldsymbol{\gamma} | \mathcal{D}) r(\boldsymbol{z} | \boldsymbol{w}, \boldsymbol{\gamma})$$

where $r(\boldsymbol{z} | \boldsymbol{w}, \boldsymbol{\gamma})$ in principle can be any distribution. We then consider the KL divergence in the extended space for $(\boldsymbol{w}, \boldsymbol{\gamma}, \boldsymbol{z})$:

$$\text{KL}\left[q(\boldsymbol{w}, \boldsymbol{\gamma}, \boldsymbol{z}) || p(\boldsymbol{w}, \boldsymbol{\gamma}, \boldsymbol{z} | \mathcal{D})\right] = \int_{\boldsymbol{z}} \sum_{\boldsymbol{\gamma}} \int_{\boldsymbol{w}} q(\boldsymbol{w}, \boldsymbol{\gamma}, \boldsymbol{z}) \log \frac{q(\boldsymbol{w}, \boldsymbol{\gamma}, \boldsymbol{z})}{p(\boldsymbol{w}, \boldsymbol{\gamma}, \boldsymbol{z} | \mathcal{D})} \, d\boldsymbol{w} d\boldsymbol{z}$$

which, by utilizing the definitions of $p(\boldsymbol{w}, \boldsymbol{\gamma}, \boldsymbol{z})$ and $q(\boldsymbol{w}, \boldsymbol{\gamma}, \boldsymbol{z})$ can be rewritten to

$$\begin{aligned}
&\text{KL}\left[q(\boldsymbol{w}, \boldsymbol{\gamma}, \boldsymbol{z}) || p(\boldsymbol{w}, \boldsymbol{\gamma}, \boldsymbol{z} | \mathcal{D})\right] \\
&= \mathbb{E}_{q(\boldsymbol{z})}\left[\text{KL}\left[q(\boldsymbol{w}, \boldsymbol{\gamma} | \boldsymbol{z}) || p(\boldsymbol{w}, \boldsymbol{\gamma})\right] + \log q(\boldsymbol{z})\right] - \mathbb{E}_{q(\boldsymbol{W}, \boldsymbol{\Gamma}, \boldsymbol{z})}\left[\log p(\mathcal{D} | \boldsymbol{w}, \boldsymbol{\gamma}) + \log r(\boldsymbol{z} | \boldsymbol{w}, \boldsymbol{\gamma})\right] + \log p(\mathcal{D}).
\end{aligned} \tag{9}$$

As shown in Ranganath et al. (2016),

$$\text{KL}\left[q(\boldsymbol{w}, \boldsymbol{\gamma}) || p(\boldsymbol{w}, \boldsymbol{\gamma} | \mathcal{D})\right] \leq \text{KL}\left[q(\boldsymbol{w}, \boldsymbol{\gamma}, \boldsymbol{z}) || p(\boldsymbol{w}, \boldsymbol{\gamma}, \boldsymbol{z} | \mathcal{D})\right], \tag{10}$$

giving a looser than the original upper bound (see Ranganath et al. (2016) for a proof), but the dependence structure in the variational posterior distribution can compensate for this.

After doing some algebra, we get the following contribution to the first term within the first expectation in Equation (9) from a specific layer:

$$\sum_{ij} \left( \tilde{\alpha}_{ij} \left( \log \frac{\sigma_{ij}}{\tilde{\sigma}_{ij}} + \log \frac{\tilde{\alpha}_{ij}}{\alpha_{ij}} - \frac{1}{2} + \frac{\tilde{\sigma}_{ij}^2 + (\tilde{\mu}_{ij} z_i - 0)^2}{2\sigma_{ij}^2} \right) + (1 - \tilde{\alpha}_{ij}) \log \frac{1 - \tilde{\alpha}_{ij}}{1 - \alpha_{ij}} \right).$$

Since we use autoregressive flows, the contribution to the second term in the first expectation simplifies to

$$\log q(\boldsymbol{z}) = \log q_0(\boldsymbol{z}_0) - \sum_{k=1}^{K} \sum_{i=1}^{n_{in}} \log \kappa_{k,i}.$$

For the specific choice of $r(\boldsymbol{z}|\boldsymbol{w}, \boldsymbol{\gamma})$, we follow Louizos et al. (2017) in choosing

$$r_B(\boldsymbol{z}_B|\boldsymbol{w}, \boldsymbol{\gamma}) = \prod_{i=1}^{n_{in}} \mathcal{N}(\nu_i, \tau_i^2).$$

We define the dependence of $\boldsymbol{\nu} = (\nu_1, ..., \nu_{in})$ and $\boldsymbol{\tau}^2 = (\tau_1^2, ..., \tau_{in}^2)$ on $\boldsymbol{w}$ and $\boldsymbol{\gamma}$ similar to Louizos & Welling (2017):

$$\begin{aligned} \boldsymbol{\nu} &= n_{out}^{-1}(\boldsymbol{d}_1 \boldsymbol{s}^T)\mathbf{1}, && \text{with } \boldsymbol{s} = \zeta(\mathbf{e}^T(\boldsymbol{w} \odot \boldsymbol{\gamma})) \\ \log \boldsymbol{\tau}^2 &= n_{out}^{-1}(\boldsymbol{d}_2 \boldsymbol{s}^T)\mathbf{1}. \end{aligned} \qquad (11)$$

Here, $\boldsymbol{d}_1$, $\boldsymbol{d}_2$ and $\mathbf{e}$ are trainable parameters with the same shape as $\boldsymbol{z}$. For $\zeta$, we use hard-tanh [1] , as opposed to tanh (used in Louizos & Welling (2017)) as this works better empirically. For the last term of Equation (9), we thus have:

$$\log r(\boldsymbol{z}|\mathbf{w}, \boldsymbol{\gamma}) = \log r_B(\boldsymbol{z}_B|\mathbf{w}, \boldsymbol{\gamma}) + \log \left| \det \frac{\partial \boldsymbol{z}_B}{\partial \boldsymbol{z}} \right|.$$

This means that we must use two normalizing flows, one to get from $\boldsymbol{z}_0$ to $\boldsymbol{z} = \boldsymbol{z}_K$, and another from $\boldsymbol{z}$ to $\boldsymbol{z}_B$. Here, we have shown the inverse normalizing flow with only one layer, but this can in general be extended to an arbitrary number of them just like in Equation (6).

For the biases of a given layer, we assume they are independent of the weights, and each other. We use the standard normal prior with the mean-field Gaussian approximate posterior. As we do not use normalizing flows on the biases, we only need to compute the KL-divergence between two Gaussian distributions:

$$\text{KL}\left[q(\boldsymbol{b})||p(\boldsymbol{b})\right] = \sum_{ij} \left( \log \frac{\sigma_{b_{ij}}}{\tilde{\sigma}_{b_{ij}}} - \frac{1}{2} + \frac{\tilde{\sigma}_{b_{ij}}^2 + (\tilde{\mu}_{b_{ij}} - 0)^2}{2\sigma_{b_{ij}}^2} \right).$$

In practice, the ELBO is optimized through a (stochastic) gradient algorithm where the reparametrization trick (Kingma & Welling, 2013) combined with mini-batch is applied. This involves sampling the large $\Gamma$ and $\mathbf{W}$ matrices.

## 4 Combining LBBNNs with the LCRT and MNF

The variational distribution in Equation (4) (used in both Hubin & Storvik (2019) and Bai et al. (2020)) has two major drawbacks when utilized in deep Bayesian neural networks. Firstly, each forward pass during training requires sampling the large $\Gamma$ and $\mathbf{W}$ matrices, consisting of all $\gamma_{ij}$'s, and $w_{ij}$'s, to compute the activations for each layer in the network, as opposed to standard BNNs that only require to sample $\mathbf{W}$. Additionally, due to the binary nature of the $\gamma_{ij}$'s, they must be approximated with a continuous distribution in order to be able to propagate gradients through them using the reparametrization trick. Here, we will show how to circumvent *both* of these issues by sampling the pre-activations $h_j$ (by which we mean the linear

---

[1]See https://pytorch.org/docs/stable/generated/torch.nn.Hardtanh.html for a definition.

combination before the non-linear activation function is applied) given in Equation (1) directly, typically referred to as the local reparametrization trick (Kingma et al., 2015, LCRT). The general idea behind the LCRT is that if we have a sum of independent Gaussian random variables, the sum will also be (exactly) Gaussian. In our variational approximation, we have a sum of independent random variables where each member of the sum is, in turn, a product of a pair of discrete and a continuous random variable. The central limit theorem still holds for a sum of independent random variables, as long as Lindeberg's condition (Billingsley, 2017) is satisfied. Thus, in order to apply the LCRT in our case, we must compute the mean and variance of the (approximate) Gaussian pre-activations: Then, we still use the same stochastic variational inference optimization algorithm as in Hubin & Storvik (2024).

$$\mathbb{E}[h_j] = \mathbb{E}\left[ b_j + \sum_{i=1}^{N} o_i \gamma_{ij} w_{ij} \right] = \tilde{\mu}_{b_j} + \sum_{i=1}^{N} o_i \tilde{\alpha}_{ij} \tilde{\mu}_{ij}$$

$$\mathrm{Var}[h_j] = \mathrm{Var}\left[ b_j + \sum_{i=1}^{N} o_i \gamma_{ij} w_{ij} \right] = \tilde{\sigma}_{b_j}^2 + \sum_{i=1}^{N} o_i^2 \tilde{\alpha}_{ij}(\tilde{\sigma}_{ij}^2 + (1 - \tilde{\alpha}_{ij})\tilde{\mu}_{ij}^2).$$

Here, $o$ denotes the output from the previous layer, consisting of $N$ neurons. Another advantage of using the LCRT is that we get a reduction in the variance of the gradient estimates, as shown in Kingma et al. (2015). Note also that the approximations induced by the sampling procedure for $\boldsymbol{h}$ also can be considered as an alternative variational approximation directly for $p(\boldsymbol{h}|\mathcal{D})$.

For our second extension, we apply normalizing flows in the activation space to increase the flexibility of the variational posterior. When using normalizing flows, the mean and the variance of the activation $h_j$ are:

$$\mathbb{E}[h_j] = \tilde{\mu}_{b_j} + \sum_{i=1}^{N} o_i z_i \tilde{\alpha}_{ij} \tilde{\mu}_{ij}$$

$$\mathrm{Var}[h_j] = \tilde{\sigma}_{b_j}^2 + \sum_{i=1}^{N} o_i^2 \tilde{\alpha}_{ij}(\tilde{\sigma}_{ij}^2 + (1 - \tilde{\alpha}_{ij})z_i^2 \tilde{\mu}_{ij}^2),$$

It should be noted that $\boldsymbol{z}$ affects both the mean and the variance of our Gaussian approximation, whereas in Louizos & Welling (2017) it only influences the mean. Louizos & Welling (2017) also sample one $\boldsymbol{z}$ for each observation within the mini-batch. We found that empirically it made no difference in performance to only sample one vector and multiply the same $\boldsymbol{z}$ with each input vector. We do this, as it is more computationally efficient.

## 5 Experiments

In this section, we demonstrate the robustness of our approach and show improvements with respect to the closest baseline methods of Hubin & Storvik (2024), (denoted LBBNN-SSP-MF in their paper), denoted LBBNN here, with the two approaches proposed in this paper. We consider both full Bayesian model averaging (Hoeting et al., 1999), averaging over the posterior distribution for the inclusion variables, as well as the use of the median probability model (Barbieri et al., 2018), only including weights with posterior probabilities for inclusion variables above 0.5. Median probability models have the potential of giving huge sparsity gains. We also compare it to other reasonable baseline methods. We are not interested in trying to obtain state-of-the-art predictive results at all costs, hence all our experiments are run *without* using ad-hoc tricks commonly found in the Bayesian deep learning literature that often improve on performance, such as tempering the posterior (Wenzel et al., 2020) or clipping the variance of the variational posterior distribution as done in Louizos & Welling (2017). Using these tricks (although tempting) would not allow us to evaluate the pure contribution of the methodology. We provide comparisons to a standard BNN, and to the multiplicative normalizing flow method (BNN-FLOW) introduced by Louizos & Welling (2017), as these are closely related to the LBBNN and its two extensions detailed in this paper. The goal is to compare results between frequentist and our Bayesian networks with the same architecture and hyper-parameters. See Figure 3 for a graphical illustration of how the different methods are related to one another. We have a

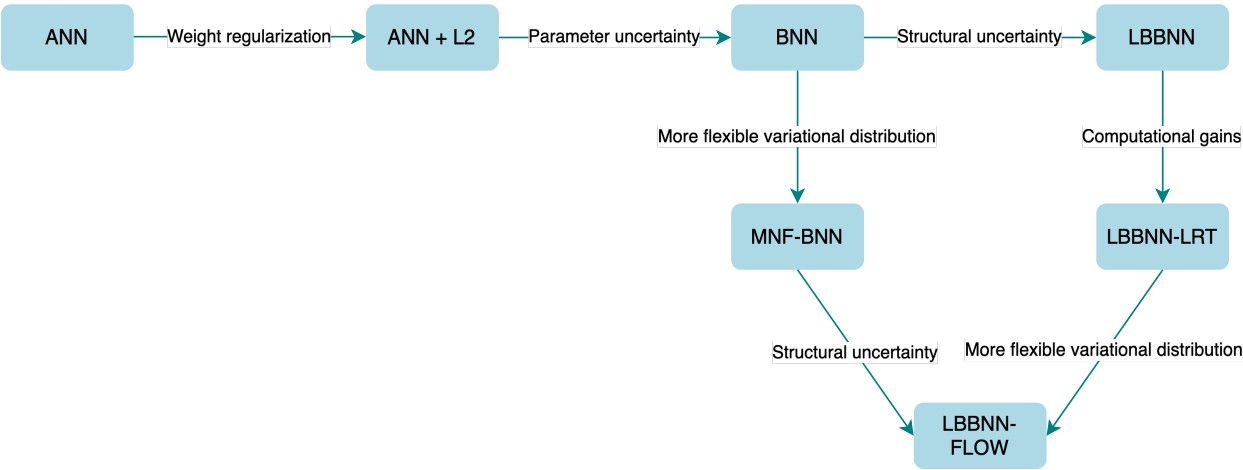

Figure 3: Illustration of the relations between the different methods considered in this paper. Exactly one design change is present between all direct neighbours disregarding the directions of the edges.

standard, frequentist neural network without any regularization (ANN), that corresponds to using maximum likelihood to estimate the weights of the network. We also have a frequentist network with L2 regularization, corresponding to the maximum a posteriori estimator (MAP) with independent Gaussian priors from a standard BNN. We added a BNN approximated with a mean-field variational inference, which we also added to comparisons. This standard BNN takes into account uncertainty in the weights, rather than finding a point estimate allowing us to evaluate the benefit of it as compared to corresponding MAP estimates. From there, we get to the LBBNN method by having an extra inclusion parameter for each weight, allowing for a sparse BNN. For LBBNN exactly the same parameter priors (slab components) as in BNN were used, allowing us to evaluate the effects of adding the structural uncertainty. The multiplicative normalizing flow method (BNN-FLOW) is also closely related to a standard BNN, but here instead of sparsifying the network, we allow the variational posterior distribution to be more flexible than a standard mean-field Gaussian, used in BNNs. Further, using the local reparametrization trick (LBBNN-LCRT) is mainly a computational advantage compared to the LBBNN method. Finally, LBBNN-FLOW (proposed in this paper) is related to *both* BNN-FLOW and LBBNN-LCRT, in the sense that it can learn a sparse BNN, and in addition have a more flexible posterior distribution than the mean-field Gaussian used in LBBNN-LCRT. In Hubin & Storvik (2019), comprehensive classification experiments show that LBBNNs can sparsify Bayesian neural networks to a large degree while maintaining high predictive power.

We demonstrate that increasing the flexibility of the variational posterior with normalizing flows improves both predictive performance and sparsity levels against the mean-field approximations for LBBNN on a set of addressed datasets. Additionally, we perform two simulation studies. In the first one, we consider variable selection in a logistic regression setting, with highly correlated explanatory variables. In the second, we generate data from clusters of two-dimensional Gaussian distributions and compare how the different methods handle predictive uncertainty. All the experiments were coded in Python, using the PyTorch deep learning library (Paszke et al., 2019). In addition to the results reported here, we also perform classification experiments on various tabular datasets, taken from the UCI machine learning repository (Kelly et al., 2023). The results (detailed in Appendix C), demonstrate that our suggested approach also works in these settings. Additionally, for the UCI datasets, for an empirical measure of calibration, we report the expected calibration error (ECE) Guo et al. (2017) on the classification problems. Further, we calculate Pinball loss (Gneiting, 2011) on regression problems, averaged across levels from 0.05 to 0.95 (see Tables 6 and 10 in AppendixC). On the regression datasets, we include two additional baselines for comparison, Gaussian Processes, using an out-of-the-box version of the package from Varvia et al. (2023), and BNNs fitted with Hamiltonian Monte Carlo (HMC), using the package from Sharaf et al. (2020). While the former performed well and on par with our methods, the latter was underperforming. Our flow method when both using the full (variational) model averaging and the median probability model is demonstrating robust performance (one may with caution

Table 1: Performance metrics on the logistic regression variable selection simulation study.

|          | CS    | LBBNN-LCRT | LBBNN-FLOW |
|----------|-------|------------|------------|
| mean TPR | 0.681 | 0.838      | **0.972**  |
| mean FPR | 0.125 | 0.084      | **0.074**  |

say even on par or better) compared to the baselines on these datasets. Finally for the UCI datasets, we additionally check the internal parsimony of all of the Bayesian baselines through the $p_{\text{WAIC}_1}$ and $p_{\text{WAIC}_2}$ penalties from Gelman et al. (2014), metrics that also have been used as estimates of the effective number of parameters.

### 5.1 Logistic regression simulation study

In this section, we do a variable selection experiment within a logistic regression setting. As logistic regression is just a special case of a neural network with one neuron (and hence one layer), modifying the algorithms is straightforward. We are limiting ourselves to the logistic regression context to be able to compare to the original baseline method from Carbonetto & Stephens (2012), who have shown that the mean-field variational approximation starts to fail the variable selection task when the covariates are correlated. As we are only interested in comparing the mean-field variational approach against the variational distribution with normalizing flows, we do not include comparisons with more traditional variable selection methods such as Lasso (Tibshirani, 1996) or Elastic Net (Zou & Hastie, 2005). We use the same data as in Hubin & Storvik (2018), consisting of a mix of 20 binary and continuous variables, with a binary outcome, and we have 2 000 observations. The covariates, $\boldsymbol{x}$, are generated with a strong and complicated correlation structure between many of the variables (see Figure 4). For more details on exactly how the covariates are generated, see appendix B of Hubin & Storvik (2018). The response variable, $y$, is generated according to the following data-generating process:

$$\eta \sim \mathcal{N}(\boldsymbol{\beta x}, 0.5)$$
$$y \sim \text{Bernoulli}\left(\frac{\exp(\eta)}{1 + \exp(\eta)}\right)$$

with the regression parameters defined to be:

$$\boldsymbol{\beta} = (-4, 0, 1, 0, 0, 0, 1, 0, 0, 0, 1.2, 0, 37.1, 0, 0, 50, -0.00005, 10, 3, 0).$$

The goal is to train the different methods to select the non-zero elements of $\boldsymbol{\beta}$. We consider the parameter $\beta_j$ to be included if the posterior inclusion probability $\alpha_j > 0.5$, i.e. the median probability model of Barbieri & Berger (2004). We fit the different methods 100 times (to the same data), each time computing the true positive rate (TPR), and the false positive rate (FPR).

In this experiment we compare our approaches LBBNN-LCRT and LBBNN-FLOW against the algorithm proposed by Carbonetto & Stephens (2012), denoted as CS henceforth. That method is very similar to LBBNN-LCRT, as it uses the same variational distribution. But in CS, optimization is done with coordinate ascent variational inference and without subsampling from the data. For the normalizing flows, we use flows of length two with the neural networks having two hidden layers of 100 neurons each. We use a batch size of 400 and train for 500 epochs. We use standard normal priors for the weights and a prior inclusion probability of 0.25 on the inclusion indicators for all three approaches. Hence, we are in the setting of a Bayesian logistic regression, with variable selection.

The results are in Table 3. We also show a bar-plot (Figure 5) for each of the 20 weights over the 100 runs. We see that LBBNN-FLOW performs best, with the highest TPR and the lowest FPR. It is especially good at picking out the correct variables where there is a high correlation between many of them (for example

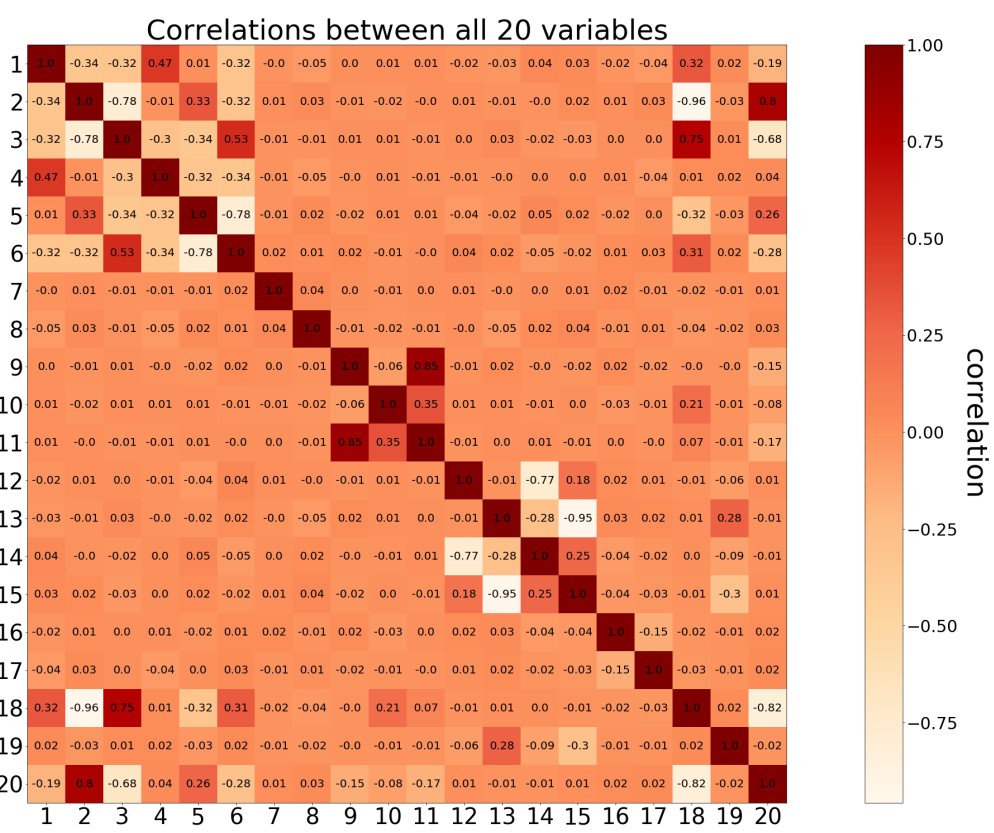

Figure 4: Plots showing the correlation between different variables in the logistic regression simulation study.

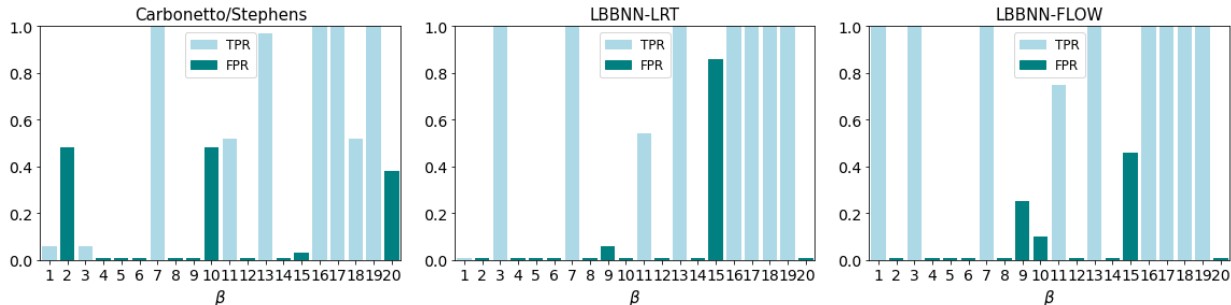

Figure 5: Bar-plots showing how often the weights are included over 100 runs.

$\beta_1 - \beta_6$). We might attribute this to the more flexible variational posterior distribution, as opposed to the mean-field Gaussian distribution used in the other three methods. Carbonetto & Stephens (2012) also discuss how the mean-field approach can only be expected to be a good approximation when the variables are independent or at most weakly correlated.

## 5.2 Predictive uncertainty

A key motivation behind using a Bayesian approach is their ability to handle predictive uncertainty more accurately than non-Bayesian neural networks. We therefore in this experiment want to illustrate how our approaches LBBNN-LCRT and LBBNN-FLOW, as well as Monte Carlo dropout (Gal & Ghahramani, 2016), and a regular (dense) BNN behave in terms of the predictive uncertainty. The purpose of this study is, thus, illustrative rather than comparative and the methods are not competing here. For this experiment, we simulate 5 clusters of data from two-dimensional Gaussian distributions. For the five Gaussians, we use the means and covariances reported in Appendix B. The data is then transformed to be in the range between 0 and 1, for ease of visualization. The task is to classify to the correct class corresponding to a specific cluster.

We generate three datasets, with 10, 50, and 200 samples from each class, respectively. For all the methods, we fit a network with one hidden layer consisting of 1000 neurons, meaning we are in a setting where the number of trainable parameters is much larger than the number of observations, which is a typical scenario for applications of Bayesian neural networks. For dropout, we use 0.5 for the dropout probability, and we use 0.5 for the prior inclusion probabilities for LBBNN-LCRT and LBBNN-FLOW. We use flows of length two, with the neural networks consisting of two hidden layers of 50 neurons each. For all the methods, we use 10 samples for model averaging. To measure predictive uncertainty, we generate a test set over a grid over $[0, 1]^2$ and compute the entropy of the predictive distributions for each point in the grid. Maximum entropy is attained when the predictive distribution is uniform, i.e. 0.2 for each class. The results are shown in Figure 6, Figure 7, and Figure 8.

With little data, we see a stark difference between dropout and the Bayesian networks. Dropout predictions are highly confident everywhere, except for at the decision boundaries between the classes. In contrast, the Bayesian networks exhibit high uncertainty in most areas, especially where little data is observed. When we increase the amount of data, we can see that the Bayesian networks gradually get more certain about predictions, and the entropies (as desired) start to converge towards the data-generative ones, while for dropout at a given rate, the uncertainties do not reduce. It should be noted that there is no under-fitting happening, as we have close to 100% accuracy during training for all the methods. As a final observation, we see that the dense BNN typically has slightly less uncertainty than LBBNN with LCRT and FLOW. However, we can not say much about how good/bad this is, since it is difficult to obtain the true uncertainties for our model, i.e. running a reversible jump MCMC (Green & Hastie, 2009) in the settings of LBBNN of a reasonable size is currently just infeasible computationally.

Additionally, we perform an experiment where we generate 10 000 test samples (2 000 from each cluster), after training with 50 samples (10 from each cluster). After training, we compute the entropy of the predictive distribution on the test data and sort the data from lowest to highest entropy. We also sort the samples based

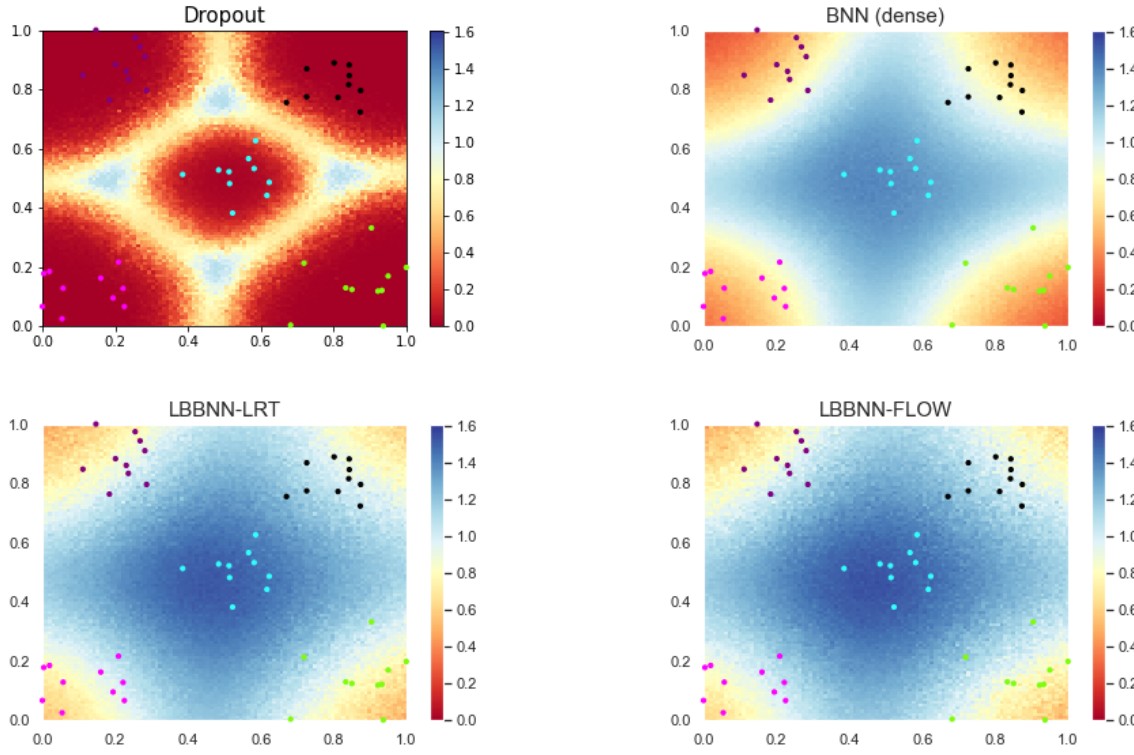

Figure 6: Entropy with 10 samples from each cluster

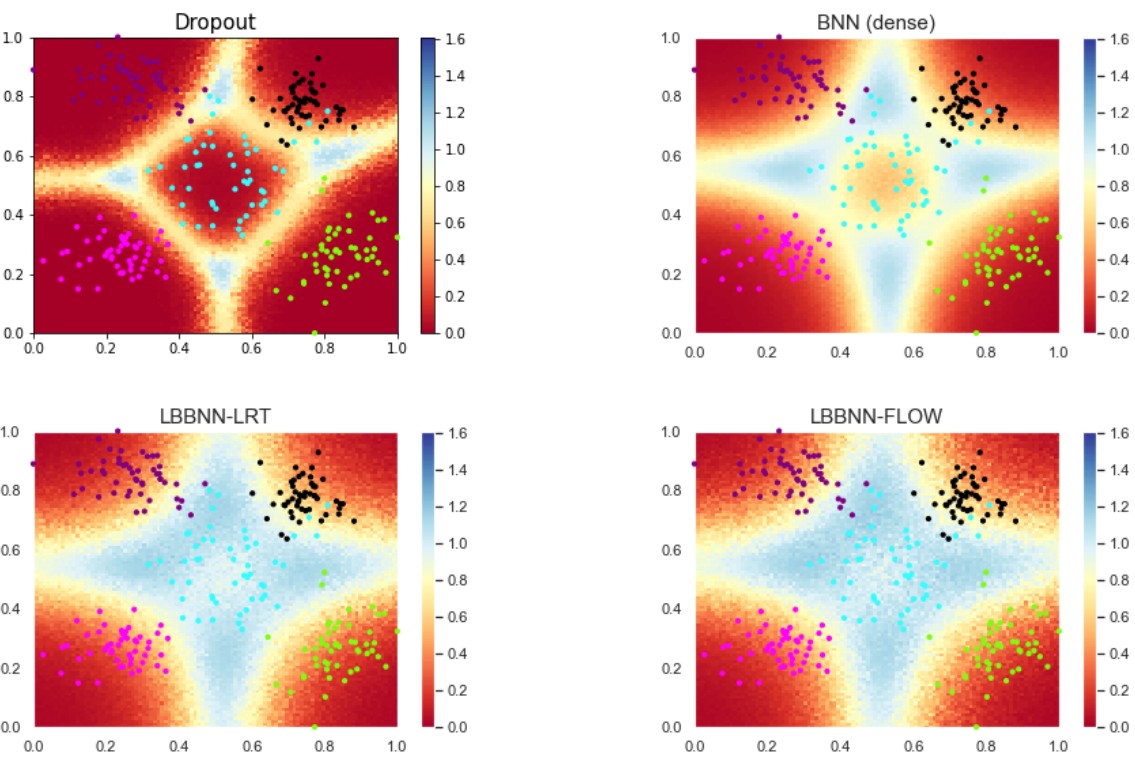

Figure 7: Entropy with 50 samples from each cluster

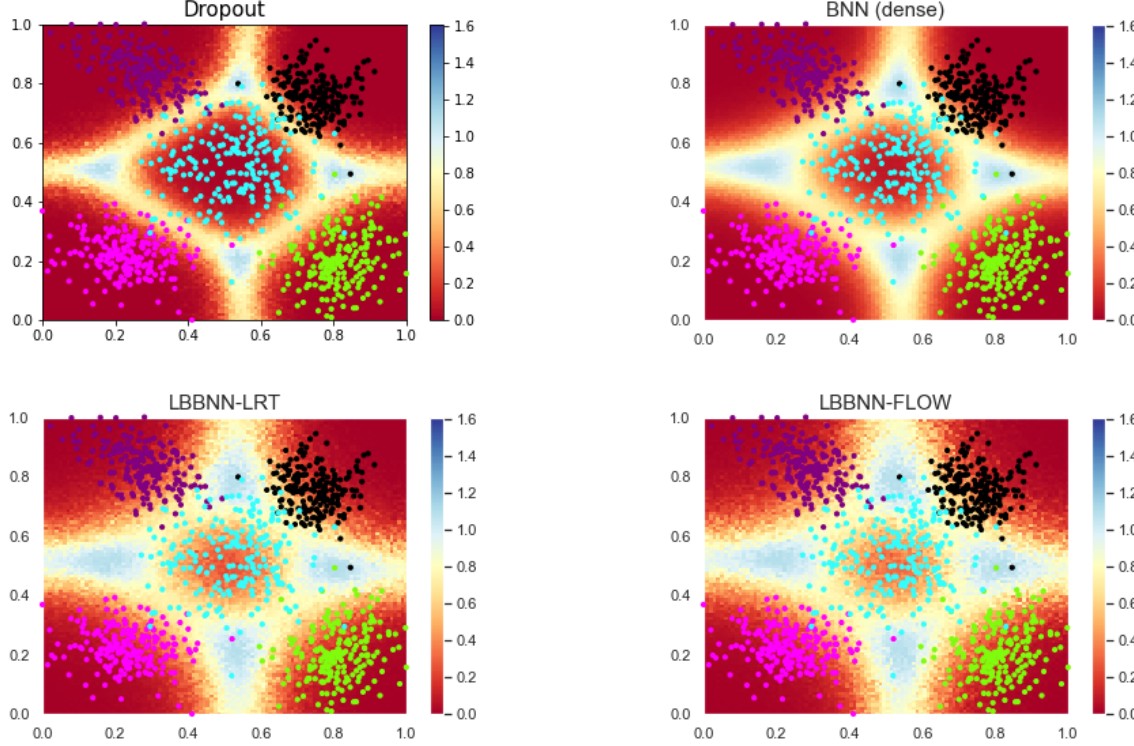

Figure 8: Entropy with 200 samples from each cluster

on the maximum class probability and compute the cumulative accuracy (with 100 data samples at a time). By that, we mean that we start with the accuracy for the 100 most confident predictions, followed by 100 less confident predictions, and so on until we reach 100 of the least confident predictions. The results are in Figure 9. With dropout, the maximum class probability is typically very high (i.e. we are extremely certain about which class the sample belongs to). After the first 5 000 (sorted) samples, the output probability for the most likely class is at around 95%. With LCRT and FLOW, on the other hand, it has dropped to roughly 50%. This mirrors what we saw earlier, dropout has high certainty most of the time. Despite this, we see that in this experiment the Bayesian methods have higher predictive accuracy than dropout for the cases with the most uncertainty.

As a final illustration, we consider an experiment where we take the maximum model averaged pre-activation output (pre-softmax) of the last layer (i.e. just before applying the softmax function) as a measure instead of using entropy. We use the training data ($m = 1000$) to generate an empirical confidence interval for the model-averaged pre-activation outputs for all the classes. We use a one-sided 95% confidence interval on the upper bound. During testing, we generate a sample over a grid, now between -1 and 2 in both dimensions, and take the highest model-averaged pre-activation output. We then check whether it falls within the empirical confidence interval or not. The results are shown in Figure 10. We see that in the regions with extremely low entropy, we can detect out-of-distribution data. This shows that using maximal entropy for out-of-distribution data as suggested in Louizos & Welling (2017) might not be optimal. However, we still see the potential of BNNs to differentiate between in and out-of-domain uncertainty using the pre-activation values of the output of BNNs. We do not go any further here and leave this topic for future research. We rather continue with some real data examples.

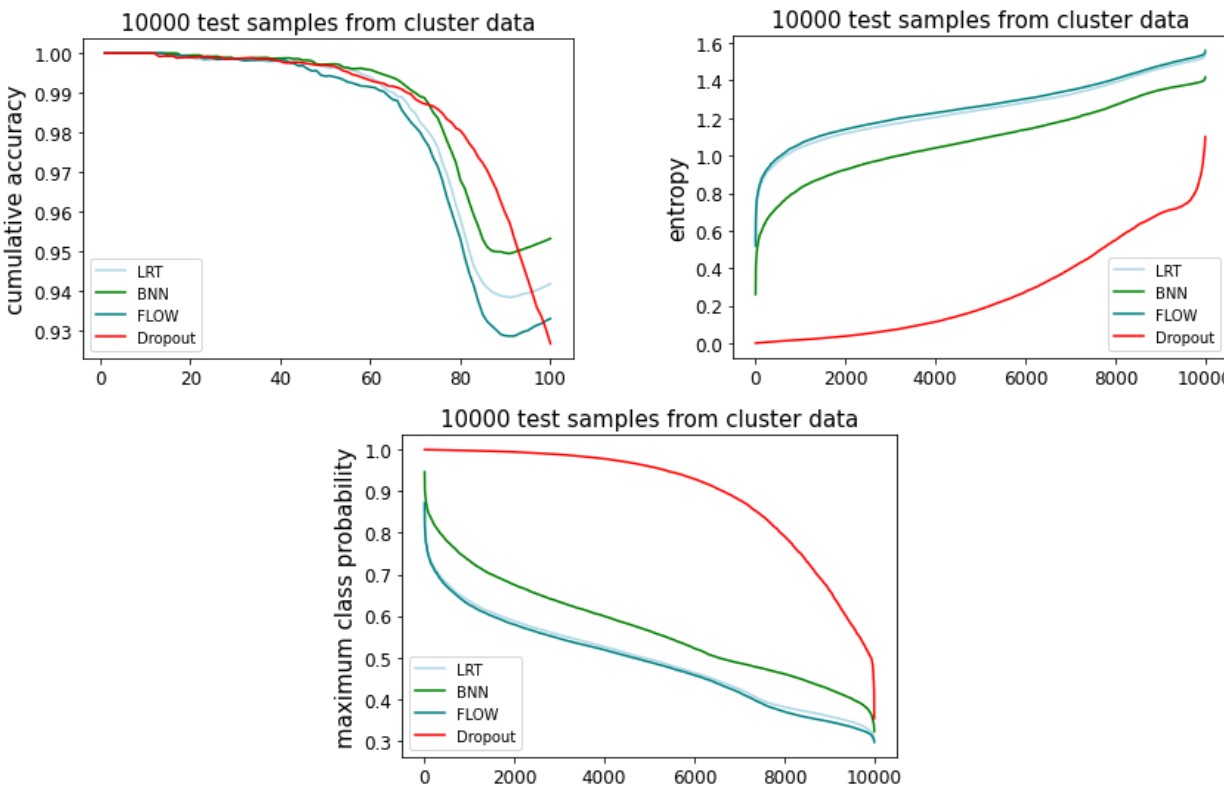

Figure 9: Top left, cumulative accuracy (100 samples at a time), where each point is the accuracy for the corresponding data points. Top right, entropy sorted from low to high. Bottom, maximum class probability sorted from high to low.

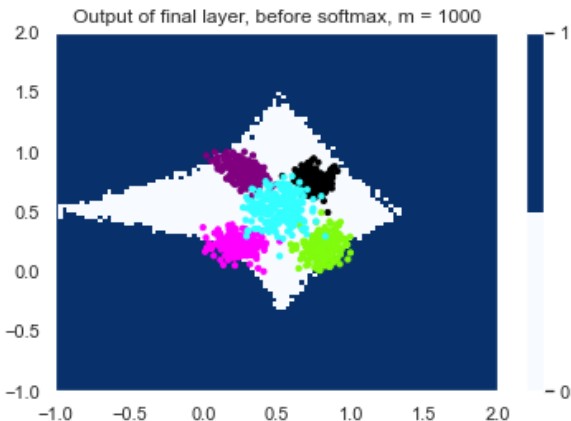

Figure 10: Out of distribution detection, where dark blue corresponds to the OOD data detected by the BNN, and white is the in-distribution data.

### 5.3 Classification experiments

We perform two classification experiments, one with the same fully connected architecture as in Hubin & Storvik (2019), and the other with a convolutional architecture (see appendix A for details on how this is implemented, while the specifications on the architecture will be provided later in the text). In both cases, we classify on MNIST (Deng, 2012), FMNIST (Fashion MNIST) (Xiao et al., 2017) and KMNIST (Kuzushiji MNIST) (Clanuwat et al., 2018). MNIST is a database of handwritten digits ranging from 0 to 9. FMNIST consists of ten different fashion items from the Zalando (Europe's largest online fashion retailer) database. Lastly, KMNIST also consists of ten classes, with each one representing one row of Hiragana, a Japanese syllabary. All of these datasets contain 28x28 grayscale images, divided into a training and validation set with 60 000 and 10 000 images respectively. MNIST and FMNIST are well-known and often utilized datasets, so it is easy to compare performance when testing novel algorithms. KMNIST is a somewhat recent addition and is considered a more challenging task than the classical MNIST digits dataset because each Hiragana can have many different symbols.

For the experiments with the fully connected architecture, we have two hidden layers with 400 and 600 neurons respectively, ReLU (Agarap, 2018) activation functions. For fitting the models, we used the Adam (Kingma & Ba, 2014) optimizer. We use a batch size of 100 and train for 250 epochs. All the experiments are run 10 times, and we report the minimum, median, and maximum predictive accuracy over these 10 runs. In addition to the performance measure (accuracy), we also report the density of the network, defined as the ratio of non-zero weights. The reported density (1-sparsity) is an average over these 10 runs. For the UCI datasets, we also measure the computed $p_{\mathrm{WAIC}_1}$, and $p_{\mathrm{WAIC}_2}$ from Gelman et al. (2014)), as these can be argued to be a measure the effective number of parameters of the models. Although, according to Gelman et al. (2014) the latter is only valid for normal linear models with large sample size, known variance, and uniform prior distribution. The results are in Tables 7, 8, and 11 in Appendix C. For BNN and BNN-FLOW, we use standard normal priors. For ANN + L2, we use the weight decay of 0.5, inducing a penalized likelihood, which corresponds to MAP (maximum aposteriori probability) solutions of BNN and BNN-FLOW under standard normal priors. For the LBBNN-LCRT and LBBNN-FLOW methods, we use the standard normal prior for the slab components of all the weights and biases in the network, and a prior inclusion probability of 0.10. For both $q(\boldsymbol{z})$ and $r(\boldsymbol{z}|\mathbf{W},\boldsymbol{\Gamma})$, we use flows of length two, where the neural networks consist of two hidden layers with 250 neurons each. For our second classification experiment, we use the LeNet-5 (LeCun et al., 1998) convolutional architecture, but with 32 and 48 filters for the convolutional layers. We use the same priors and normalizing flows as in the previous experiment, and the same datasets. We emphasize that it is possible to use deeper and more complicated architectures (for example Resnet-18 (He et al., 2016)), which may improve the results reported in this paper. As the goal here is not to try to approach (or *hack* through tuning and engineering) state-of-the-art results, we do not experiment any further with this. To

Table 2: Performance metrics (accuracy and density) on the KMNIST, MNIST, FMNIST validation data, for the fully connected architecture. For the accuracies (%), we report the minimum, maximum, and median over the ten different runs. Density is computed as an average over the ten runs. The best median results are bold.

| **KMNIST** | Median probability model | | | | Full model averaging | | | |
|---|---|---|---|---|---|---|---|---|
| Method | min | median | max | density | min | median | max | density |
| LBBNN | 89.22 | 89.59 | 89.98 | 0.113 | 89.43 | 89.76 | 90.21 | 1.000 |
| LBBNN-LCRT | 90.04 | 90.26 | 90.43 | 0.136 | 90.23 | 90.39 | 90.60 | 1.000 |
| LBBNN-FLOW | 90.64 | **91.12** | 91.46 | **0.096** | 91.16 | 91.30 | 91.61 | 1.000 |
| BNN-FLOW | - | - | - | - | 92.02 | 92.28 | 92.61 | 1.000 |
| BNN | - | - | - | - | 92.21 | **92.53** | 92.64 | 1.000 |
| ANN | - | - | - | - | 90.44 | 91.02 | 91.28 | 1.000 |
| ANN + L2 | - | - | - | - | 87.24 | 87.76 | 88.15 | 1.000 |
| **MNIST** | Median probability model | | | | Full model averaging | | | |
| Method | min | median | max | density | min | median | max | density |
| LBBNN | 98.01 | 98.10 | 98.20 | 0.098 | 98.03 | 98.14 | 98.23 | 1.000 |
| LBBNN-LCRT | 97.84 | 97.95 | 98.09 | 0.103 | 98.01 | 98.08 | 98.11 | 1.000 |
| LBBNN-FLOW | 98.14 | **98.36** | 98.42 | **0.074** | 98.23 | 98.42 | 98.53 | 1.000 |
| BNN-FLOW | - | - | - | - | 98.43 | **98.58** | 98.63 | 1.000 |
| BNN | - | - | - | - | 98.36 | 98.48 | 98.63 | 1.000 |
| ANN | - | - | - | - | 97.95 | 98.13 | 98.20 | 1.000 |
| ANN + L2 | - | - | - | - | 96.97 | 97.05 | 97.16 | 1.000 |
| **FMNIST** | Median probability model | | | | Full model averaging | | | |
| Method | min | median | max | density | min | median | max | density |
| LBBNN | 88.47 | 88.76 | 88.90 | 0.106 | 88.60 | 88.74 | 88.91 | 1.000 |
| LBBNN-LCRT | 87.51 | 87.82 | 87.94 | 0.141 | 87.88 | 87.94 | 88.14 | 1.000 |
| LBBNN-FLOW | 89.49 | **89.70** | 89.88 | **0.097** | 89.52 | 89.80 | 89.92 | 1.000 |
| BNN-FLOW | - | - | - | - | 89.19 | 89.42 | 89.53 | 1.000 |
| BNN | - | - | - | - | 90.07 | **90.20** | 90.43 | 1.000 |
| ANN | - | - | - | - | 88.75 | 89.51 | 89.88 | 1.000 |
| ANN + L2 | - | - | - | - | 86.85 | 87.37 | 87.54 | 1.000 |

measure predictive performance, we consider two approaches. First, the fully variational Bayesian model averaging approach, where we average over 100 samples from the variational posterior distribution, taking into account uncertainty in both weights and structures following Hubin & Storvik (2019). Secondly, we consider the median probability model (Barbieri & Berger, 2004), where we only do model averaging over the weights that have a posterior inclusion probability greater than 0.5, whilst others are excluded from the model. This allows for significant sparsification of the network. We emphasize that this is possible because we can go back to sampling the weights when doing inference, i.e. we sample only from the weights that have a corresponding inclusion probability greater than 0.5. We also report the density, i.e. the proportion of weights included in the median probability model. For the full variational model averaging approach we consider the density to be equal to one since we do not explicitly exclude any weights when computing the predictions (even though a large proportion of the weights may have a small inclusion probability and in practice within any 10 samples over which we are marginalizing less than 100% of the weights will be used, yet ideally one wants to average over more than 10 samples).

The results with the fully connected architecture can be found in Table 2 and for the convolutional architecture in Table 3. Firstly, we see that using the LBBNN-LCRT gives results that are comparable to the baseline LBBNN method, except for FMNIST where it performs a bit worse both with the fully connected

Table 3: Performance metrics on the KMNIST, MNIST, FMNIST validation data, with the convolutional architecture. See the caption in Table 2 for more details.

| **KMNIST** | Median probability model | | | | Full model averaging | | | |
|---|---|---|---|---|---|---|---|---|
| Method | min | median | max | density | min | median | max | density |
| LBBNN | 95.13 | 95.52 | 95.89 | 0.359 | 95.21 | 95.48 | 95.78 | 1.000 |
| LBBNN-LCRT | 94.73 | 94.94 | 95.16 | 0.429 | 95.07 | 95.42 | 95.65 | 1.000 |
| LBBNN-FLOW | 95.73 | **95.99** | 96.43 | **0.351** | 96.00 | 96.18 | 96.44 | 1.000 |
| BNN-FLOW | - | - | - | - | 96.14 | **96.42** | 96.64 | 1.000 |
| BNN | - | - | - | - | 95.19 | 95.34 | 95.58 | 1.000 |
| ANN | - | - | - | - | 94.18 | 94.95 | 95.27 | 1.000 |
| ANN + L2 | - | - | - | - | 92.00 | 92.51 | 92.77 | 1.000 |

| **MNIST** | Median probability model | | | | Full model averaging | | | |
|---|---|---|---|---|---|---|---|---|
| Method | min | median | max | density | min | median | max | density |
| LBBNN | 99.22 | 99.26 | 99.35 | 0.353 | 99.21 | 99.28 | 99.33 | 1.000 |
| LBBNN-LCRT | 99.11 | 99.26 | 99.31 | 0.406 | 99.20 | 99.28 | 99.34 | 1.000 |
| LBBNN-FLOW | 99.15 | **99.27** | 99.41 | **0.338** | 99.16 | 99.29 | 99.42 | 1.000 |
| BNN-FLOW | - | - | - | - | 99.26 | **99.32** | 99.41 | 1.000 |
| BNN | - | - | - | - | 99.21 | 99.30 | 99.36 | 1.000 |
| ANN | - | - | - | - | 99.01 | 99.15 | 99.23 | 1.000 |
| ANN + L2 | - | - | - | - | 97.93 | 98.30 | 98.40 | 1.000 |

| **FMNIST** | Median probability model | | | | Full model averaging | | | |
|---|---|---|---|---|---|---|---|---|
| Method | min | median | max | density | min | median | max | density |
| LBBNN | 91.14 | 91.31 | 91.48 | **0.352** | 91.10 | 91.26 | 91.44 | 1.000 |
| LBBNN-LCRT | 90.04 | 90.40 | 90.85 | 0.433 | 90.52 | 90.73 | 91.06 | 1.000 |
| LBBNN-FLOW | 90.52 | **91.54** | 91.75 | 0.367 | 91.38 | 91.71 | 92.04 | 1.000 |
| BNN-FLOW | - | - | - | - | 91.60 | **91.87** | 92.10 | 1.000 |
| BNN | - | - | - | - | 91.04 | 91.60 | 91.99 | 1.000 |
| ANN | - | - | - | - | 90.40 | 91.21 | 91.63 | 1.000 |
| ANN + L2 | - | - | - | - | 87.79 | 88.05 | 88.48 | 1.000 |

and with the convolutional architecture. It is no surprise that these results are similar, as using the LCRT is mainly a computational advantage. Secondly, we note that the LBBNN-FLOW method performs better than the other two methods, on both convolutional and fully connected architectures, while having the most sparse networks. We also see that LBBNN-FLOW performs well compared to the BNN and BNN-FLOW architectures, especially on the fully connected architecture where it gets comparable accuracy even with very sparse networks. The higher density in general on the convolutional architectures is mainly a result of them being already sparse in the beginning. However, these networks could also be sparsified further by using more conservative priors on inclusions of the weights. The increased predictive power of using normalizing flows comes at a computational cost. With the fully connected architecture, we observed that it took around 4 seconds to train one epoch with LBBNN-LCRT, 13 seconds with LBBNN, and 17 seconds with LBBNN-FLOW on an NVIDIA A10 GPU. On the convolutional architecture, it took 7 seconds per epoch with the LBBNN-LCRT, 18 seconds with LBBNN, and 28 with normalizing flows. We note that the frequentist networks perform slightly worse on these datasets with our chosen architectures. The results could likely be improved by adding more regularization, such as dropout or batch-normalization, but we do not do this here, as we are not interested in trying to obtain state-of-the-art results. Naturally, the frequentist networks are much more computationally efficient, as they only have half the parameters of a standard BNN.

## 6 Discussion

We have demonstrated that increasing the flexibility in the variational posterior distribution with normalizing flows improves the predictive power compared to the baseline method (with mean-field posterior) while obtaining more sparse networks, despite having a looser variational bound than the mean-field approach. Also, the flow method performed best on a variable selection problem demonstrating better structure learning performance, while the mean-field approaches struggle with highly correlated variables. More generally, we argue that Bayesian neural networks (BNNs) are much better at obtaining realistic predictive uncertainty estimates than their frequentist counterparts, as they have higher uncertainty when data is sparse. We do not observe a big difference in the uncertainty estimates obtained with dense BNN compared to our approaches. Also, calibration of uncertainties in predictive applications is similar with a slight advantage of the proposed in this paper approach. Unlike dense BNNs, our methods have the additional advantage of being able to perform variable selection. The downside is that LBBNNs have an extra parameter per weight, making them less computationally efficient than dense BNNs. Using normalizing flows is a further computational burden as we must also optimize over all the extra flow parameters. If uncertainty handling is not desirable, one could gain the minimal number of predictive parameters using the model trained with flows by relying on the posterior means of the median probability model's parameters. This approach is studied for simpler approximations in more detail in Hubin & Storvik (2024) but it is omitted in this paper.

In this paper, we use the same prior for all the weights and inclusion indicators, although this is not necessary. A possible avenue of further research could be to vary the prior inclusion probabilities, to induce different sparsity structures or to incorporate the actual prior knowledge about prior inclusion probabilities of the covariates. Currently, we are taking into account uncertainty in weights and parameters, given some neural network architecture. In the future, it may be of interest to see if it is also possible to incorporate uncertainty in the activation functions. By having skip connections to the output, we could learn with uncertainties to skip all non-linear layers if a linear function is enough, or if a constant estimate of the parameters of the responses is enough (null model), or if one needs some nonlinear layers. This could lead to more transparent Bayesian deep learning models. But the success in that task relies on sufficiently good structure learning, where mean field-approximations are known to not work well (Carbonetto & Stephens, 2012).

A possible application is to do a genome-wide association study (GWAS), using our method. Combining LBBNNs and GWAS has been proposed by Demetci et al. (2021), however, this only uses the mean-field posterior. With our normalizing flow approach, we can easily model dependencies within each SNP set, in addition to dependencies between the different SNP sets. Other set of promising applications are recovering structural equations in nonlinear dynamical systems as the robust and uncertainty aware alternative to $l1$ penalty used in the Sindy approach (Brunton et al., 2016).

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

## Supplementary material

The code used for the experiments can be found in the accompanying zip folder.

## A Convolutional architectures

For convolutional layers, the variational distribution is defined to be:

$$q_{\boldsymbol{\theta}}(\mathbf{W}|\boldsymbol{\Gamma}, \boldsymbol{z}) = \prod_{i=1}^{n_h}\prod_{j=1}^{n_w}\prod_{k=1}^{n_f}[\gamma_{ijk}\mathcal{N}(w_{ijk}; z_k\tilde{\mu}_{ijk}, \tilde{\sigma}_{ijk}^2) + (1 - \gamma_{ijk})\delta(w_{ijk})] \tag{12}$$

$$q_{\tilde{\alpha}_{ijk}}(\gamma_{ijk}) = \text{Bernoulli}(\gamma_{ijk}; \tilde{\alpha}_{ijk}),$$

where $n_h$, $n_w$, and $n_f$ denote the height, width, and number of filters in the convolutional kernel.

For the convolutional layers, we use the following for the inverse normalizing flows:

$$\boldsymbol{\nu} = ((\text{Mat}(\boldsymbol{W} \odot \boldsymbol{\Gamma})\mathbf{e}) \otimes \mathbf{d}_1)(\mathbf{1} \odot (n_{\text{h}}n_{\text{w}})^{-1})$$

$$\log \boldsymbol{\tau^2} = ((\text{Mat}(\boldsymbol{W} \odot \boldsymbol{\Gamma})\mathbf{e}) \otimes \mathbf{d}_2)(\mathbf{1} \odot (n_{\text{h}}n_{\text{w}})^{-1}). \tag{13}$$

Here, $\text{Mat}(\cdot)$ denotes the matricisation operator (as defined in Louizos & Welling (2017)), i.e. changing the shape of a multidimensional tensor into a matrix.

## B Data for predictive uncertainty experiments

For the predictive uncertainty experiment, we generate data from the following Gaussian distributions:

$$G_1 \sim \mathcal{N}\left(\begin{pmatrix} -8 \\ -8 \end{pmatrix}, \begin{pmatrix} 6 & -1 \\ -1 & 3.5 \end{pmatrix}\right),$$

$$G_2 \sim \mathcal{N}\left(\begin{pmatrix} 6 \\ 6 \end{pmatrix}, \begin{pmatrix} 0 & 3 \\ 3 & 0 \end{pmatrix}\right),$$

$$G_3 \sim \mathcal{N}\left(\begin{pmatrix} -7 \\ 8 \end{pmatrix}, \begin{pmatrix} -3 & 4 \\ -5 & 1 \end{pmatrix}\right),$$

$$G_4 \sim \mathcal{N}\left(\begin{pmatrix} 8 \\ -8 \end{pmatrix}, \begin{pmatrix} 0 & 5 \\ 4 & 2 \end{pmatrix}\right),$$

$$G_5 \sim \mathcal{N}\left(\begin{pmatrix} 0 \\ 0 \end{pmatrix}, \begin{pmatrix} 0 & 9 \\ 9 & 0 \end{pmatrix}\right).$$

## C Experiments on tabular datasets

In this experiment, we consider six classification datasets and three regression datasets. We compare our approach LBBNN-FLOW against LBBNN-LCRT, LBBNN, a dense BNN, BNN-FLOW, ANN+L2, ANN, and Monte Carlo (MC) dropout (with dropout rates corresponding to our prior inclusion probabilities), in addition to Gaussian processes and Hamiltonian Monte Carlo for the regression datasets. For the neural networks (except those trained with HMC), we use a single hidden layer with 500 neurons and again train for 250 epochs with the Adam optimizer. For HMC, we use the default tuning parameters from Sharaf et al. (2020) run single chains for 100000 iterations. We also use 10-fold cross-validation, and report the minimum, mean and maximum accuracy over these 10 repetitions, in addition to the mean sparsity. We also report the expected calibration error (ECE), in addition to $p_{\text{WAIC}_1}$ and $p_{\text{WAIC}_2}$ for the classification datasets, while for the regression datasets we report RMSE, the pinball loss, $p_{\text{WAIC}_1}$ and $p_{\text{WAIC}_2}$.

For the six classification datasets, most are taken from the UCI machine learning repository. The Credit Approval dataset (Quinlan, 2007) consists of 690 samples with 15 variables, with the response variable being whether someone gets approved for a credit card or not. The Bank Marketing dataset (Moro et al., 2012) consists of data (45211 samples and 17 variables) related to a marketing campaign of a Portuguese banking institution, where the goal is to classify whether the persons subscribed to the service or not. In addition to this, we use the Census Income dataset (Kohavi, 1996) with 48842 samples and 14 variables, where we try to classify whether someone's income exceeds 50000 dollars per year. Additionally, we have three datasets

related to classifying food items. The first, the Raisins dataset (ÇINAR et al., 2020), consists of 900 samples and 7 variables, where the goal is to classify into two different types of raisings grown in Turkey. Secondly, we use the Dry Beans dataset (UCI, 2020), consisting of 13611 samples, 17 variables, and 7 different types of beans. Lastly, the Pistachio dataset (Ozkan et al., 2021) consists of 2148 samples and 28 variables, with two different types of Pistachios.

For the regression datasets, we use the abalone shell data (Nash,Warwick, Sellers,Tracy, Talbot,Simon, Cawthorn,Andrew, and Ford,Wes, 1995), the wine quality dataset (Cortez,Paulo, Cerdeira,A., Almeida,F., Matos,T., and Reis,J., 2009), and the Boston housing data (Harrison Jr & Rubinfeld, 1978). To avoid model misspecification, responses were standardized for the regression datasets. The variance of the responses was then assumed fixed and equal to 1 in all of the models, while the mean parameter was modeled.

For the classification datasets, we see that our LBBNN-FLOW method typically performs well compared to the LBBNN baseline, mostly having higher predictive power, also with the median probability model. It also performs well compared to our other baseline methods. On calibration, we see that our method again performs well compared to baselines. Finally, for the $p_{\text{WAIC}_1}$ and $p_{\text{WAIC}_2}$, we see that using the median probability model typically reduces this metric. We note that our method often has lower values than BNN-FLOW and BNN, even though these methods have fewer parameters. As mentioned before, however, it is unclear how good of an estimate this is for the effective number of parameters in highly non-linear methods such as neural networks. For the regression examples, our method performs slightly worse than some baselines in terms of RMSE and pinball loss, however, LBBNN-LCRT performs best (on average) on one dataset. Here, it is worth noting that convergence checks for HMC were not satisfactory not allowing us to claim the algorithms have converged. Tuning the parameters of the sampler and increasing the number of chains were not helpful in this case.

Table 4: Performance results on the Credit Approval, Bank Marketing, and Census Income datasets, using 10-fold cross-validation. The minimum, mean and maximum accuracies are reported, in addition to the density. The best results are bold.

| Credit Approval | Median probability model | | | | Full model averaging | | | |
|---|---|---|---|---|---|---|---|---|
| Method | min | mean | max | density | min | mean | max | density |
| LBBNN | 81.16 | 85.22 | 91.30 | 0.431 | 81.16 | 85.80 | 91.30 | 1.000 |
| LBBNN-LCRT | 81.16 | 86.23 | 92.75 | 0.347 | 81.16 | 86.23 | 91.30 | 1.000 |
| LBBNN-FLOW | 84.10 | **88.55** | 94.20 | 0.348 | 82.61 | **87.68** | 91.30 | 1.000 |
| BNN-FLOW | - | - | - | - | 82.61 | 86.23 | 89.86 | 1.000 |
| BNN | - | - | - | - | 78.26 | 83.33 | 88.41 | 1.000 |
| ANN | - | - | - | - | 78.26 | 83.19 | 91.30 | 1.000 |
| ANN + L2 | - | - | - | - | 73.91 | 83.19 | 89.86 | 1.000 |
| ANN + MC dropout | - | - | - | - | 81.16 | 86.52 | 92.75 | 1.000 |

| Bank Marketing | Median probability model | | | | Full model averaging | | | |
|---|---|---|---|---|---|---|---|---|
| Method | min | mean | max | density | min | mean | max | density |
| LBBNN | 89.75 | 90.66 | 91.74 | 0.430 | 89.75 | 90.61 | 91.43 | 1.000 |
| LBBNN-LCRT | 90.75 | 91.27 | 92.16 | 0.347 | 90.60 | 91.27 | 92.16 | 1.000 |
| LBBNN-FLOW | 90.58 | **91.38** | 92.08 | 0.347 | 90.75 | **91.36** | 92.03 | 1.000 |
| BNN-FLOW | - | - | - | - | 90.14 | 91.13 | 91.96 | 1.000 |
| BNN | - | - | - | - | 90.63 | 91.16 | 91.74 | 1.000 |
| ANN | - | - | - | - | 90.93 | 90.97 | 91.62 | 1.000 |
| ANN + L2 | - | - | - | - | 90.75 | 91.15 | 91.77 | 1.000 |
| ANN + MC dropout | - | - | - | - | 90.70 | 91.17 | 92.06 | 1.000 |

| Cencus Income | Median probability model | | | | Full model averaging | | | |
|---|---|---|---|---|---|---|---|---|
| Method | min | mean | max | density | min | mean | max | density |
| LBBNN | 85.24 | 85.74 | 86.49 | 0.431 | 85.52 | 85.85 | 86.69 | 1.000 |
| LBBNN-LCRT | 85.36 | 85.90 | 86.77 | 0.349 | 85.63 | 85.92 | 86.57 | 1.000 |
| LBBNN-FLOW | 85.60 | **86.04** | 86.43 | 0.349 | 85.57 | **86.05** | 86.49 | 1.000 |
| BNN-FLOW | - | - | - | - | 85.05 | 85.32 | 85.79 | 1.000 |
| BNN | - | - | - | - | 84.77 | 85.27 | 86.45 | 1.000 |
| ANN | - | - | - | - | 84.56 | 85.09 | 85.54 | 1.000 |
| ANN + L2 | - | - | - | - | 84.73 | 85.39 | 85.91 | 1.000 |
| ANN + MC dropout | - | - | - | - | 85.32 | 85.89 | 86.49 | 1.000 |

Table 5: Performance results on the Dry Beans, Pistachio, and Raisin datasets, using 10-fold cross-validation. The minimum, mean and maximum accuracies are reported, in addition to the density. The best results are bold.

| **Dry Beans** | Median probability model | | | | Full model averaging | | | |
|---|---|---|---|---|---|---|---|---|
| Method | min | mean | max | density | min | mean | max | density |
| LBBNN | 90.88 | 92.65 | 93.90 | 0.442 | 91.25 | 92.80 | 93.82 | 1.000 |
| LBBNN-LCRT | 91.62 | **93.18** | 94.41 | 0.349 | 91.69 | 93.34 | 94.34 | 1.000 |
| LBBNN-FLOW | 89.26 | 92.38 | 93.90 | 0.279 | 89.85 | 92.57 | 94.19 | 1.000 |
| BNN-FLOW | - | - | - | - | 91.32 | 93.05 | 94.19 | 1.000 |
| BNN | - | - | - | - | 91.47 | 93.35 | 94.63 | 1.000 |
| ANN | - | - | - | - | 91.47 | 93.37 | 94.71 | 1.000 |
| ANN + L2 | - | - | - | - | 91.54 | **93.38** | 94.71 | 1.000 |
| ANN + MC dropout | - | - | - | - | 91.40 | 92.96 | 94.04 | 1.000 |

| **Pistachio** | Median probability model | | | | Full model averaging | | | |
|---|---|---|---|---|---|---|---|---|
| Method | min | mean | max | density | min | mean | max | density |
| LBBNN | 91.12 | 93.46 | 96.26 | 0.433 | 91.12 | 93.36 | 95.33 | 1.000 |
| LBBNN-LCRT | 91.59 | **93.93** | 95.79 | 0.350 | 92.06 | 94.07 | 95.79 | 1.000 |
| LBBNN-FLOW | 91.12 | 93.46 | 95.33 | 0.350 | 91.12 | 93.46 | 95.33 | 1.000 |
| BNN-FLOW | - | - | - | - | 90.65 | 93.60 | 96.26 | 1.000 |
| BNN | - | - | - | - | 92.52 | 94.07 | 96.26 | 1.000 |
| ANN | - | - | - | - | 92.06 | 94.11 | 96.73 | 1.000 |
| ANN + L2 | - | - | - | - | 92.06 | 93.93 | 96.26 | 1.000 |
| ANN + MC dropout | - | - | - | - | 91.12 | **94.16** | 96.26 | 1.000 |

| **Raisins** | Median probability model | | | | Full model averaging | | | |
|---|---|---|---|---|---|---|---|---|
| Method | min | mean | max | density | min | mean | max | density |
| LBBNN | 83.33 | **87.00** | 92.22 | 0.439 | 83.33 | 86.78 | 92.22 | 1.000 |
| LBBNN-LCRT | 81.11 | 86.11 | 91.11 | 0.349 | 81.11 | 86.78 | 92.22 | 1.000 |
| LBBNN-FLOW | 83.33 | 86.67 | 92.22 | 0.349 | 82.22 | 86.56 | 92.22 | 1.000 |
| BNN-FLOW | - | - | - | - | 82.22 | 87.22 | 91.11 | 1.000 |
| BNN | - | - | - | - | 81.11 | **87.89** | 92.22 | 1.000 |
| ANN | - | - | - | - | 81.11 | 86.44 | 90.00 | 1.000 |
| ANN + L2 | - | - | - | - | 81.11 | 87.56 | 92.22 | 1.000 |
| ANN + MC dropout | - | - | - | - | 81.11 | 87.00 | 93.33 | 1.000 |

Table 6: Expected calibration error, with minimum, mean, and maximum values obtained using 10-fold cross-validation. MPM denotes the medium probability model.

| Method | ECE (min, mean, max) | | | | | |
|---|---|---|---|---|---|---|
| | Credit Approval | Bank Marketing | Census Income | Dry Beans | Pistachio | Raisins |
| LBBNN | (0.056, 0.079, 0.127) | (0.023, 0.031, 0.039) | (0.005, **0.010**, 0.015) | (0.011, 0.016, 0.021) | (0.012, 0.028, 0.047) | (0.043, 0.073, 0.097) |
| LBBNN-MPM | (0.035, 0.080, 0.122) | (0.013, 0.025, 0.033) | (0.006, 0.011, 0.016) | (0.002, 0.015, 0.031 ) | (0.014, 0.028, 0.043) | (0.035, 0.069, 0.103) |
| LBBNN-LCRT | (0.035, 0.071,0.097) | (0.001, 0.016, 0.020) | (0.006, **0.010**, 0.012) | (0.008, 0.014, 0.019) | (0.014, **0.022**, 0.034) | (0.034, 0.070, 0.094) |
| LBBNN-LCRT-MPM | (0.015, 0.073, 0.119) | (0.011, 0.015, 0.018) | (0.008, 0.012, 0.016) | (0.007, **0.013**, 0.025) | (0.012, 0.025, 0.035) | (0.057, 0.076, 0.115) |
| LBBNN-FLOW | (0.030, **0.069**, 0.103) | (0.002, **0.007**, 0.015) | (0.007, 0.011, 0.016) | (0.010, 0.017, 0.040) | (0.011, 0.024, 0.045) | (0.018, 0.068, 0.103) |
| LBBNN-FLOW-MPM | (0.025, 0.071, 0.126) | (0.003, 0.008, 0.012) | (0.009, 0.012, 0.015 ) | (0.008, 0.014, 0.026) | (0.012, 0.030, 0.047) | (0.038, 0.072, 0.096) |
| BNN-FLOW | (0.025, 0.074, 0.123) | (0.006, 0.012, 0.022) | (0.020, 0.027, 0.047) | (0.007, 0.014, 0.029) | (0.010, 0.029, 0.055) | (0.038, 0.069, 0.104) |
| BNN | (0.046, 0.097, 0.195) | (0.010, 0.015, 0.024) | (0.020, 0.025, 0.031) | (0.005, **0.013**, 0.030) | (0.010, 0.039, 0.057) | (0.034, 0.070, 0.103) |
| ANN | (0.072, 0.131, 0.188) | (0.012, 0.017, 0.023) | (0.020, 0.025, 0.031) | (0.007, **0.013**, 0.021) | (0.027, 0.042, 0.056) | (0.040, 0.074, 0.109) |
| ANN + L2 | (0.020, 0.114, 0.178) | (0.013, 0.018, 0.024) | (0.015, 0.020, 0.026) | (0.007, **0.013**, 0.020) | (0.014, 0.040, 0.053) | (0.025, **0.066**, 0.111) |
| ANN + MC dropout | (0.028, 0.079, 0.115) | (0.011, 0.014, 0.017) | (0.008, 0.011, 0.018) | (0.033, 0.039, 0.048) | (0.016, 0.029, 0.051) | (0.027, 0.069, 0.113) |

Table 7: The $p_{\mathrm{WAIC}_1}$ metric, on our six classification datasets, where the minimum, mean, and maximum values are obtained with 10-fold cross-validation.

| | $p_{\mathrm{WAIC}_1}$ (min, mean, max) | | | | | |
|---|---|---|---|---|---|---|
| Method | Credit Approval | Bank Marketing | Census Income | Dry Beans | Pistachio | Raisins |
| LBBNN | (0.620, 1.323, 2.595) | (28.18, 31.15, 34,74) | (22.98, 29.33, 37.14) | (15.82, 23.18, 30.44) | (2.594, 3.277, 4.762) | (0.149, 0.271, 0.668) |
| **LBBNN-MPM** | **(0.000, 0.000, 0.000)** | **(0.001, 0.002, 0.003)** | **(0.002, 0.002, 0.003)** | **(0.001, 0.002, 0.003)** | **(0.000, 0.000, 0.000)** | **(0.000, 0.000, 0.000)** |
| LBBNN-LCRT | (0.013, 0.026, 0.041) | (0.575, 0.643, 0.725) | (0.997, 1.144, 1.357) | (9.937, 13.32, 20.12) | (0.015, 0.034, 0.052) | (0.011, 0.023, 0.043) |
| LBBNN-LCRT-MPM | (0.011, 0.025, 0.043) | (0.566, 0.645, 0.718) | (1.013, 1.140, 1.180) | (0.002, 0.002, 0.003) | (0.016, 0.032, 0.054) | (0.011, 0.024, 0.045) |
| LBBNN-FLOW | (0.009, 0.021, 0.033) | (0.486, 0.542, 0.608) | (1.045, 1.216, 1.488) | (12.83, 17.66, 26.65) | (0.016, 0.034, 0.058) | (0.011, 0.025, 0.052) |
| LBBNN-FLOW-MPM | (0.009, 0.022, 0.037) | (0.476, 0.544, 0.621) | (1.065, 1.245, 1.501) | (0.331, 0.420, 0.567) | (0.013, 0.033, 0.055) | (0.011, 0.026, 0.055) |
| BNN-FLOW | (0.012, 0.029, 0.054) | (0.523, 0.593, 0.655) | (1.235, 1.398, 1.745) | (0.402, 0.599, 1.089) | (0.017, 0.039, 0.062) | (0.012, 0.023, 0.045) |
| BNN | (0.015, 0.055, 0.255) | (0.580, 0.686, 0.760) | (1.165, 1.352, 1.442) | (0.005, 0.007, 0.009) | (0.019, 0.048, 0.073) | (0.011, 0.022, 0.034) |
| MC dropout | (0.016, 0.058, 0.249) | (0.536, 0.600, 0.687) | (1.057, 1.201, 1.353) | (75.99, 105.3, 133.2) | (0.027, 0.040, 0.059) | (0.011, 0.027, 0.042) |

Table 8: The $p_{\mathrm{WAIC}_2}$ metric, on our six classification datasets, where the minimum, mean, and maximum values are obtained with 10-fold cross-validation.

| | $p_{\mathrm{WAIC}_2}$ (min, mean, max) | | | | | |
|---|---|---|---|---|---|---|
| Method | Credit Approval | Bank Marketing | Census Income | Dry Beans | Pistachio | Raisins |
| LBBNN | (0.649, 1.367, 2.711) | (28.18, 31.73, 35,74) | (23.20, 65.55, 396.7) | (16.69, 24.55, 32.75) | (2.724, 3.485, 5.058) | (0.150, 0.269, 0.647) |
| **LBBNN-MPM** | **(0.000, 0.000, 0.000)** | **(0.001, 0.002, 0.003)** | **(0.002, 0.002, 0.003)** | **(0.001 0.002, 0.003)** | **(0.000, 0.000, 0.000)** | **(0.000, 0.000, 0.000)** |
| LBBNN-LCRT | (0.021, 0.058, 0.111) | (1.016, 1.176, 1.437) | (1.588, 2.981, 12.04) | (10.38, 13.86, 20.86) | (0.023, 0.078, 0.138) | (0.015, 0.044, 0.101) |
| LBBNN-LCRT-MPM | (0.017, 0.056, 0.114) | (0.985, 1.188, 1.374) | (1.1639, 2.041, 2.247) | (0.002, 0.002, 0.003) | (0.026, 0.073, 0.151) | (0.016, 0.047, 0.112) |
| LBBNN-FLOW | (0.014, 0.043, 0.072) | (0.780, 0.926, 1.154) | (1.718, 7.906, 21.91) | (13.30, 18.88, 29.37) | (0.025, 0.071, 0.137) | (0.016, 0.058, 0.197) |
| LBBNN-FLOW-MPM | (0.014, 0.047, 0.083) | (0.755, 0.934, 1.193) | (1.781, 8.940, 21.97) | (0.331, 0.421, 0.569) | (0.018, 0.070, 0.126) | (0.016, 0.059, 0.188) |
| BNN-FLOW | (0.020, 0.074, 0.199) | (0.856, 1.049, 1.303) | (2.230, 4.745, 12.83) | (0.403, 0.601, 1.093) | (0.030, 0.097, 0.178) | (0.016, 0.046, 0.123) |
| BNN | (0.024, 1.087, 10.14) | (1.036, 1.307, 1.517) | (2.052, 3.591, 12.34) | (0.005, 0.007, 0.009) | (0.034, 0.137, 0.244) | (0.015, 0.039, 0.064) |
| MC dropout | (0.027, 1.112, 10.13) | (0.905, 1.059, 1.351) | (1.767, 4.148, 12.03) | (101.0, 148.4, 265.8) | (0.054, 0.102, 0.216) | (0.015, 0.064, 0.181) |

Table 9: Root mean squared error, using 10-fold cross-validation. The minimum, mean and maximum are reported, in addition to the density. Best results are bold.

| Abalone | Median probability model | | | | Full model averaging | | | |
|---|---|---|---|---|---|---|---|---|
| Method | min | mean | max | density | min | mean | max | density |
| LBBNN | 0.597 | 0.677 | 0.799 | 0.435 | 0.577 | 0.665 | 0.795 | 1.000 |
| LBBNN-LCRT | 0.564 | **0.644** | 0.736 | 0.350 | 0.560 | **0.641** | 0.722 | 1.000 |
| LBBNN-FLOW | 0.577 | 0.660 | 0.767 | 0.350 | 0.572 | 0.657 | 0.761 | 1.000 |
| BNN-FLOW | - | - | - | - | 0.585 | 0.654 | 0.733 | 1.000 |
| BNN | - | - | - | - | 0.579 | 0.651 | 0.759 | 1.000 |
| ANN | - | - | - | - | 0.575 | 0.657 | 0.801 | 1.000 |
| ANN + L2 | - | - | - | - | 0.572 | 0.652 | 0.764 | 1.000 |
| ANN + MC dropout | - | - | - | - | 0.579 | 0.655 | 0.759 | 1.000 |
| Gaussian process | - | - | - | - | 0.570 | 0.650 | 0.734 | 1.000 |
| BNN-HMC | - | - | - | - | 1.047 | 1.316 | 1.778 | 1.000 |

| Wine Quality | Median probability model | | | | Full model averaging | | | |
|---|---|---|---|---|---|---|---|---|
| Method | min | mean | max | density | min | mean | max | density |
| LBBNN | 0.768 | 0.805 | 0.854 | 0.435 | 0.757 | 0.799 | 0.845 | 1.000 |
| LBBNN-LCRT | 0.742 | **0.782** | 0.820 | 0.350 | 0.741 | 0.780 | 0.820 | 1.000 |
| LBBNN-FLOW | 0.751 | 0.788 | 0.822 | 0.351 | 0.750 | 0.787 | 0.823 | 1.000 |
| BNN-FLOW | - | - | - | - | 0.747 | 0.778 | 0.815 | 1.000 |
| BNN-HMC | - | - | - | - | 0.749 | 0.767 | 0.790 | 1.000 |
| ANN | - | - | - | - | 0.740 | 0.761 | 0.792 | 1.000 |
| ANN + L2 | - | - | - | - | 0.746 | 0.763 | 0.789 | 1.000 |
| ANN + MC dropout | - | - | - | - | 0.758 | 0.799 | 0.847 | 1.000 |
| Gaussian process | - | - | - | - | 0.700 | **0.739** | 0.777 | 1.000 |
| BNN-HMC | - | - | - | - | 0.937 | 1.172 | 1.451 | 1.000 |

| Boston Housing | Median probability model | | | | Full model averaging | | | |
|---|---|---|---|---|---|---|---|---|
| Method | min | mean | max | density | min | mean | max | density |
| LBBNN | 0.267 | 0.412 | 0.621 | 0.431 | 0.262 | 0.393 | 0.552 | 1.000 |
| LBBNN-LCRT | 0.245 | **0.374** | 0.560 | 0.350 | 0.233 | 0.366 | 0.534 | 1.000 |
| LBBNN-FLOW | 0.267 | 0.402 | 0.595 | 0.352 | 0.253 | 0.395 | 0.561 | 1.000 |
| BNN-FLOW | - | - | - | - | 0.241 | 0.363 | 0.524 | 1.000 |
| BNN | - | - | - | - | 0.240 | 0.351 | 0.481 | 1.000 |
| ANN | - | - | - | - | 0.247 | 0.364 | 0.597 | 1.000 |
| ANN + L2 | - | - | - | - | 0.234 | **0.346** | 0.487 | 1.000 |
| ANN + MC dropout | - | - | - | - | 0.244 | 0.371 | 0.502 | 1.000 |
| Gaussian process | - | - | - | - | 0.217 | 0.349 | 0.444 | 1.000 |
| BNN-HMC | - | - | - | - | 0.611 | 1.014 | 1.320 | 1.000 |

Table 10: Mean pinball loss on a grid between 0.05 and 0.95 in increments of 0.05. Min, mean, and max values were obtained using 10-fold cross-validation.

| | Mean pinball (min, mean, max) | | |
|---|---|---|---|
| Method | Abalone | Wine Quality | Boston Housing |
| LBBNN | (0.210, 0.236, 0.260) | (0.291, 0.310, 0.325) | (0.100, 0.133, 0.164) |
| LBBNN-MPM | (0.220, 0.242, 0.260) | (0.296, 0.313, 0.328) | (0.097, 0.137, 0.167) |
| LBBNN-LCRT | (0.203, **0.227**, 0.251) | (0.289, 0.304, 0.316) | (0.087, 0.121, 0.150) |
| LBBNN-LCRT-MPM | (0.204, 0.228, 0.259) | (0.290, 0.304, 0.318) | (0.089, 0.123, 0.151) |
| LBBNN-FLOW | (0.208, 0.232, 0.253) | (0.289 0.306 0.321) | (0.091, 0.132, 0.158) |
| LBBNN-FLOW-MPM | (0.209, 0.231, 0.261) | (0.290, 0.306 0.319) | (0.092, 0.133, 0.154) |
| MNF | (0.211, 0.232, 0.250) | (0.293, 0.303, 0.317) | (0.085, 0.120, 0.142) |
| BNN | (0.210, 0.230, 0.255) | (0.286, 0.295, 0.307) | (0.084, 0.117, 0.145) |
| ANN | (0.210, 0.231, 0.255) | (0.279, 0.292, 0.301) | (0.087, 0.119, 0.149) |
| ANN + L2 | (0.208, 0.230, 0.255) | (0.283, 0.293, 0.301) | (0.083, **0.114**, 0.142) |
| ANN + MC dropout | (0.211, 0.231, 0.252) | (0.295, 0.312, 0.328) | (0.089, 0.125, 0.156) |
| Gaussian process | (0.210, 0.231, 0.252) | (0.267, **0.283**, 0.295) | (0.079, 0.117, 0.145) |
| BNN-HMC | (0.358, 0.522, 0.749) | (0.368, 0.464, 0.597) | (0.223, 0.389, 0.503) |

Table 11: The $p_{\mathrm{WAIC}_1}$ and $p_{\mathrm{WAIC}_2}$ metric, on our three regression datasets, where the minimum, mean and maximum values are obtained with 10-fold cross-validation.

| | $p_{\mathrm{WAIC}_1}$ (min, mean, max) | | | $p_{\mathrm{WAIC}_2}$ (min, mean, max) | | |
|---|---|---|---|---|---|---|
| Method | Abalone | Wine Quality | Boston Housing | Abalone | Wine Quality | Boston Housing |
| LBBNN | (1.937, 4.486, 14.09) | (6.304, 7.496, 8.617) | (0.037, 0.202, 0.890) | (1.987, 5.206, 20.01) | (6.660, 7.832, 9.642) | (0.038, 0.210, 0.933) |
| LBBNN-MPM | **(0.000, 0.001, 0.002)** | **(0.001, 0.001, 0.003)** | **(0.000, 0.000, 0.000)** | **(0.000, 0.001, 0.002)** | **(0.001, 0.001, 0.003)** | **(0.000, 0.000, 0.000)** |
| LBBNN-LCRT | (0.766, 1.469, 2.228) | (4.978, 5.620, 6.318) | (0.013, 0.128, 0.483) | (0.774, 1.528, 2.595) | (5.112, 5.888, 6.989) | (0.013, 0.133, 0.514) |
| LBBNN-LCRT-MPM | (0.001, 0.001, 0.003) | (0.003, 0.004, 0.006) | **(0.000, 0.000, 0.000)** | (0.001, 0.001, 0.003) | (0.003, 0.004, 0.006) | **(0.000, 0.000, 0.000)** |
| LBBNN-FLOW | (0.563, 1.923, 7.515) | (3.220, 4.202, 6.069) | (0.025, 0.223, 1.025) | (0.568, 2.191, 9.925) | (3.273, 4.407, 7.352) | (0.025, 0.238, 1.157) |
| LBBNN-FLOW-MPM | (0.039, 0.147, 0.430) | (0.171, 0.226, 0.268) | (0.001, 0.012, 0.077) | (0.039, 0.147, 0.432) | (0.172, 0.226, 0.269) | (0.001, 0.012, 0.078) |
| BNN-FLOW | (0.074, 0.237, 0.468) | (0.362, 0.609, 2.210) | (0.003, 0.021, 0.062) | (0.075, 0.238, 0.473) | (0.363, 0.632, 2.415) | (0.003, 0.021, 0.062) |
| BNN | (0.002, 0.007, 0.045) | (0.014, 0.019, 0.039) | **(0.000, 0.000, 0.000)** | (0.002, 0.007, 0.045) | (0.014, 0.019, 0.039) | **(0.000, 0.000, 0.000)** |
| MC dropout | (7.295, 15.42, 44.78) | (18.70, 22.81, 27.87) | (0.344, 1.721, 6.331) | (7.972, 56.82, 430.4) | (20.35, 29.71, 71.86) | (0.397, 2.658, 12.43) |

