# OpenReview forum: "Sparsifying Bayesian neural networks with latent binary variables and normalizing flows"
_TMLR — Rejected by TMLR_

### Review · Reviewer_A9N5 · 2024-02-18

**Summary Of Contributions:**

The authors combine several existing ideas to extend Bayesian NNs (BNNs): spike-&-slab priors on the weights for sparsity, normalizing flows to represent posterior dependencies, and the "local reparameterization trick" which moves the variational approximation from the weights to the pre-activations. They present results on MNIST and derivatives, and synthetic logistic regression problems.

**Audience:**

Yes

**Claims And Evidence:**

Yes

**Requested Changes:**

Look at calibration on the classification datasets they already use, and ideally in some regression settings too.
Edit for length, clarity around what exactly z is, and order.

**Strengths And Weaknesses:**

The paper is reasonably well written although probably a little longer than it needs to be and could be edited down. The order is also odd in places: the simulated data examples coming before the MNIST(+) experiments, and mentioning the LRT long before describing it. The introduction of z was a bit jarring to me: is z a model parameter (in which case it should be described before the inference is described), or just an auxiliary variable for q (in which case that should be said explicitly). What is the domain of z? (the reals clearly but that isn't obvious when it's first introduced).

Combining existing ideas obviously feels somewhat incremental, but is a good fit for TMLR. My main concern with the manuscript is that uncertainty/calibration is only assessed on simulated data, and mostly on the final Gaussian clusters data, which is very simple (only two dimensions in particular). It would be more valuable to assess calibration (e.g. via ECE) on real world data (and ideally regression problems not just classification). This is the main contribution one would hope for from a sophisticated BNNs: I am OK with the authors' argument that outright predictive accuracy shouldn't be the primary goal, but my impression is existing BNNs do no better than simpler approaches (dropout, deep ensembles) for predictive uncertainty and still require post-hoc calibration. I have no idea from the paper whether these extensions are helping address this.

Minor things:
- to me (and anyone who has done much classical stats) LRT=likelihood ratio test, so this is unfortunate acronym.
- I disagree with the claim that BNNs incorporate prior knowledge. Maybe they could, but they are not used in that way.
- I don't think mean field BNNs should be called "fully Bayesian" since they are only representing a small amount of local uncertainty
- I think you should be comparing to dropout at least (especially when checking calibration)
- for the logistic regression example it would be valuable assess whether the NF based q correctly models that the correlated features should result in posterior correlation between the corresponding weights
- cut the definition of TPR and FPR, we know what those are

---

> ### Author Response · Authors · 2024-03-25
> **Response to Reviewer 1 (A9N5)**
>
> We thank the reviewer for constructive feedback that allowed us to improve the paper. We have now revised the manuscript. The updated text is in purple.
> We have changed the order of the experiments starting with 2 simulation studies, then followed by image classification. The tabular data experiments are kept in the appendix, where additionally we added 3 regression experiments on Abalone, Wine Quality, and Boston Housing datasets. Also, ECE is additionally computed for all UCI classification problems (table 6), where for 5 of 6 datasets the lowest mean ECE across 10 folds is achieved by one of the proposed inference approaches to LBBNN.  Although the minimums and maximums are overlapping a lot and we can not claim statistically significant improvements, we observe reasonable and robust calibration with potential improvement. This is an important sanity check, but we think Interpretation about “better-worse” should be done with caution as we advocate in the answer to reviewer 3.
> Also, for the 3 regression datasets (that we added to the revised paper), we reported RMSE (table 9) and for an  additional calibration check, the Pinball loss (table 10). There is no clear winner: for Abalone the best average performance is achieved by the LBBNN-LCRT (LRT is changed to LCRT along the paper to avoid confusion with likelihood ratio tests), for Wine quality by gaussian process regressions (GP), which are added to respond to the suggestion of reviewer 3, and for Boston housing by ANN + L2, however just as for classification there are a lot of overlaps across min and max summaries across the folds. Worth mentioning, however, is that dropout performs worse than LBBNNs on average in terms of calibration in almost all experiments. Yet, again, caution should be practiced. .
> As you mentioned, z  is defined as an auxiliary random variable, with the domain being the real numbers. We have clarified this in section 3.2 of the revised paper. We also have rewritten 3.2 in order to make it appear less jarring. We hope the description is now improved.
> Regarding other minor comments:
>  1. LRT is changed to LCRT to avoid confusion with the likelihood ratio test, we also refer to the paper that proposed it first in the introduction.
>  2. We completely agree that incorporating real prior knowledge is uncommon in BNNs, and in most  applications convenience priors are used (i.e. independent standard Gaussians). There are some works that incorporate prior knowledge through “pretraining” the priors like in Sam et al. (2024)  and Tran et al. (2022)  to incorporate relevant knowledge. For our priors, a researcher can adjust prior inclusion probabilities for the covariates, the common approach used in Bayesian model selection, if prior knowledge about the inclusion probabilities is available, however we did not incorporate such knowledge in our experiments, where prior inclusions are used to control sparsity in BNN applications and correspond to uniform model priors for the logistic regression simulation study.
> 3. We agree that variational BNNs cannot be called fully Bayesian, we have corrected this to fully variational Bayesian along the text.
> 4. Regarding the variable selection experiment, here we were interested in selecting the true effects and not selecting false effects, including the covariates correlated to the true effect, which is challenging. But ideally, we remove them from the model, thus in MPM the post selected effect of the false effect correlated to the true effects should be 0. And hence we concentrate on variable selection metrics.
> 5. The definitions of TPR and FPR are cut from the paper, which will save some space.
> B.-H. Tran, S. Rossi, D. Milios, and M. Filippone. All you need is a good functional prior for Bayesian deep learning. Journal of Machine Learning Research, 23(74): 1–56, 2022.
> D. Sam, R. Pukdee, D. P. Jeong, Y. Byun, and J. Z. Kolter. Bayesian neural networks with domain knowledge priors. arXiv preprint arXiv:2402.13410, 2024.

---

### Review · Reviewer_WWin · 2024-03-08

**Summary Of Contributions:**

The paper motivates, builds, makes efficient, and trains a sparse bayesian neural network.  To do this it uses a spike and slab prior, as well as a spike and slap posterior with some structural sharing between weights.  The shared structural latents are themselves fit with a flow, and the whole model is fit variationally with the local reparameterization trick.

**Audience:**

Yes

**Broader Impact Concerns:**

No concerns.

**Claims And Evidence:**

Yes

**Requested Changes:**

I would expand the discussion of the local reparameterization trick, I was lost upon first reading how it was used in the LBBNN.

I would also like you to comment more on how the full model average had full density, even if the individual or mean predictions were sparse.  This seems to run afoul of the usual stories of sparsity, in that many of the weights are unecessary, here it seems instead that all weights are utilized, just in different elements of the ensemble.

I'd be curious to see some attempt to estimate the effective dimensionality of the networks, perhaps by way of a WAIC (the Widely Applicable Information Criterion of Watanabe) style estimate, i.e. I'd be curious about the gap between the predictive likelihood and the average likelihood, i.e.:

$$  2  \sum_{i=1}^n \left( \log E_{post}[ p(y_i|x_i,\theta) ] - E_{post}[ \log p(y_i|x_i,\theta) ]  \right) $$

Which we ought to be able to interpret as an effective number of parameters.  I'd be curious if this was similar to the observed densities for the individual samples.

**Strengths And Weaknesses:**

This is a straight forward idea to try, took some clever tricks to make tractable and the authors built it and tested it on MNIST KMNIST, and FMNIST.

In this case, the straightforwardness of the idea I think is a strength.  Obviously, testing something these days on the MNIST family of models doesn't provide a huge amount of evidence of superiority of a method, but I think in this case the goal was less to claim that these LBBNN networks were superior to alternatives, rather that they can be built and show good performance, and later show qualitatively good predictive uncertainty.

---

> ### Author Response · Authors · 2024-03-25
> **Response to Reviewer 2 (WWin)**
>
> We thank the reviewer for the thoughtful and constructive feedback that allowed us to improve the paper. The important changes in the revised paper are marked in purple. Below, we shall respond to the raised comments.
> In regards to the questions/comments:
>
> 1. We have rewritten section 4 regarding the local reparametrization trick and how it is used in our case. Hopefully this is more clear now.
>
> 2. The reason we say that full model averaging has density = 1 is because we use all the weights for computing the Bayesian model average, even though many of them may have an inclusion probability near 0. In practice within 10 samples (that we used for BMA) less that 100% of weights are used. Yet as we increase the number of samples more and more weights will be used at least once, for that reason we cannot claim real sparsification in full (variational) model averaging. However, we incorporate into this process model (structural) uncertainties. Structural uncertainties incorporated allow us to do model selection using for example the median probability model (a model selection criterion like WAIC could also be addressed at this stage, but we left it outside the scope of this paper). When we use the median probability model (MPM), we remove all weights with a posterior inclusion probability less than 0.5 and only integrate out weights over the remaining ones when computing the posterior predictive. For MPM, we achieve quite significant sparsification. We have updated the text to be more clear about this.
>
> 3. We compute pWAIC1 and pWAIC2 (equation 11), from Gelman et. al (2014), to see if this can be used as an estimate for the effective number of parameters. See table 7 and 8 for classification and table 11 for regression datasets. We notice some interesting patterns. Firstly, it seems that using the median probability model in many cases indeed reduces the values of pWAIC1 and pWAIC2, especially for LBBNNs. Secondly, on the classification datasets, our flow method typically has lower values of pWAICs than BNN and BNN-FLOWs. On the regression datasets, it is BNNs that have the lowest values. However, in almost all cases, the flow method has lower values than Monte Carlo dropout. Thus the pWAICs seem to be somewhat correlated with parsimony in our case. However, we also note that it might be unclear how well these metrics work for highly non-linear models like neural networks. As the penalty is derived for simpler models, i.e. following Gelman et. al (2014), “For a normal linear model with large sample size, known variance, and uniform prior distribution on the coefficients, pWAIC1 and pWAIC2 are approximately equal to the number of parameters in the model.” Meaning that the penalties estimating the effective number of parameters might need to be adjusted to neural networks opening a very interesting direction for a potentially sound model selection criterion for BNNs.
>
> Gelman, Andrew, Jessica Hwang, and Aki Vehtari. "Understanding predictive information criteria for Bayesian models." Statistics and computing 24 (2014): 997-1016.

---

### Review · Reviewer_K8JC · 2024-03-11

**Summary Of Contributions:**

The authors propose modeling and optimization improvements to latent binary Bayesian neural networks (LBBNN): Bayesian neural networks with a spike-and-slab prior and a variational spike-and-slab posterior on the weights. The authors propose using a conditionally factorized posterior approximation instead of a fully factorized one to improve the expressivity of the variational posterior and use a normalizing flow to model the posterior distribution of the newly introduced conditioning variable. They perform some toy experiments on MNIST-type datasets and synthetic data.

**Audience:**

No

**Claims And Evidence:**

Yes

**Requested Changes:**

For me to recommend acceptance, the authors would need to give reasonable motivation for why their particular model is useful for the TMLR community. In particular, there isn't much justification for why sparsity is desirable in any of the experiments conducted by the authors, except for the variable selection experiment.

Furthermore, I think the authors should compare the calibration of the predictive uncertainty to that of standard baselines, such as Gaussian processes and HMC-sampled BNNs.

As for the writing, the introduction should be rewritten because, except for the first few sentences, it is mostly a literature review/ related works section rather than an introduction. Furthermore, I would suggest the authors clearly lay out their contributions at the end of the introduction section as a bullet-pointed list.

Finally, I have some miscellaneous points:
- What is $f$ above eq 1? Why is $\eta(x)$ called the mean vector?
- There's a sum over $k$ missing in the eq above eq 9
 - hard-tanh is not standard terminology; please define it in the text, e.g., as a footnote.
 - What do the square brackets mean in the sum in the first equation in section 4?
 - The brackets used for the expectation and variance in section 4 are inconsistent, sometimes round, sometimes square.
 - Most figures, especially Figure 3, are blurry and pixelated; please change them to vector graphics.
 - Switching from BNN to MNF in Table 1 is misleading; an "MNF" is also a BNN. Hence, I suggest the authors change it to MNF-BNN.
 - What is $M$ in the TPR and FPR definitions?

**Strengths And Weaknesses:**

## Strengths
The authors' description of how they are extending earlier models and inference procedures in Sections 2-4 is clear and easy to follow. Furthermore, they also perform extensive ablation studies, so it is easy to see what improvement is brought about by each of the authors' proposed extensions.

## Weaknesses
I have two main issues with the paper:
 - The authors' contribution seems to be very incremental
 - The use cases of the authors' method are unclear

The author's two technical contributions are combining two previous models (LBBNNs and flow posteriors) and deriving a local reparameterization trick (LRT) for their model. However, the combination is straightforward and requires no new technical insight. Similarly, the experimental methodology is correct, but the experiments are standard and do not seem to give any new insight. Overall, I do not think this work would be of interest to the TMLR community.

The above ties in with my second problem with the paper: I could not find much motivation for why the authors developed the model. In particular, it is unclear why inducing sparsity in the model weights is useful, especially given that the new model requires many new parameters for the normalizing flow used in the variational posterior. The authors also do not study the calibration of the model's uncertainty estimates very thoroughly, as they do not compare with relevant methods such as HMC + Gibbs sampling the structure or using a Laplace approximation for the approximate posterior. Furthermore, looking at Figures 6-8, it appears that the authors' proposed model yields uncertainty estimates whose calibration is even worse than the mean-field Gaussian ones, as the predictive entropy for the cyan class is a lot higher than I would expect.

---

> ### Author Response · Authors · 2024-03-25
> **Response to Reviewer 3 (K8JC). Part 1, other parts to follow.**
>
> We thank the reviewer for valuable feedback and appreciate the constructive comments that allowed us to improve the paper. We also thank the reviewer for positive feedback on the claims and evidence. In the revised manuscript, all major changes are highlighted in purple. Below are our responses to the reviewers concerns:
>
> Comment: The authors' contribution seems to be very incremental
>
> Our response: We understand the reviewer’s concern: we have an adaptation and combination of two existing techniques to a new methodological application. And while we cannot increase the objective level of novelty in this paper, we rewrote the introduction with enhanced motivation to make it more clear that we need structural uncertainty and  better approximations for structural uncertainty in BNNs. There, we also emphasize that sparsity and model selection is desired as it can lead to on one hand more transparent models, but also requiring less memory and compute for predictions, which is important for devices with limited memory to motivate additional use cases. Also discussion why more complicated variational approximations are useful and that one can incorporate knowledge through prior inclusion probabilities (which can lead to more interpretable results) are added in the paper. Contributions are now clearly stated out at the end of the introduction. We hope it will be of interest to at least some parts of the TMLR community.
>
> Comment: The use cases of the authors' method are unclear
>
> Our response: We believe that our method has several possible use cases, e.g. in situations where sparsity and transparency are desirable and features may be highly correlated. We demonstrated variable selection properties improving over a seminal work published in Bayesian analysis (Carbonetto and Stephens, 2012), who showed the mean field fails under the correlated data regime. In our study, the data was highly correlated, yet the suggested method with flows showed robust performance improving the mean-field. We believe our method could work well in high dimensional problems where one wants to do variable selection, and the solution is likely to be sparse. (e.g. genome wide association studies as in the aforementioned article), and where MCMC is too computationally costly.
> Another possible use case is as a Bayesian alternative to SINDy, a model discovery method for nonlinear dynamical systems  using sparse regression (typically LASSO). While several Bayesian alternatives exist, none of them take into account structural uncertainty, and we believe a flexible variational posterior distribution would be important in this case as well. Also, learning even more fully the structure of BNNs with allowed skipped connections for more interpretable sparse BNNs  is an interesting avenue that we now pursue in the follow up research, which however requires the foundations from this paper. These thoughts are added to the Discussion section of the paper.
>
> Question: Why is sparsity useful?
>
> Our response: For us, parsimony is the goal here that reduces prediction and saving costs, with the hope for more transparent and explainable models in the future. As mentioned, we demonstrated variable selection properties improving over a seminal work on variational methods published in Bayesian analysis (Carbonetto and Stephens, 2012). We believe it is important to demonstrate that our flow method can also remove a similar or higher amount of weights in BNNs compared to the LBBNN method, with good predictive power. But most importantly, we are interested in sparsity because of the use cases where sparse solutions are desirable, and we require a scalable inference method that can also take into account correlated weights.
> Following the advice of the second reviewer, we added pWAICs for estimating the effective numbers of parameters for all the UCI experiments, where the median probability model demonstrates reductions in the estimates of pWAICs. But, of course, flows would have larger pWAICs than LCRT or the original LBBNN, this is the price we pay for modeling dependencies (that said caution should be practiced for pWAICs for BNN as we discuss in the response to the Reviewer 2). We hope this gives additional justifications. Also for the flows, one could use posterior mean based models as in Hubin and Storvik (2024), allowing for maximal reduction of effective number of parameters, but that would disregard predictive uncertainties. We added this discussion to the Discussion section of the paper.

---

> > ### Author Response · Authors · 2024-03-25
> > **Response to Reviewer 3 (K8JC). Part 2, other parts to follow.**
> >
> > Question: Do we get good calibration?
> >
> > More generally around calibration, a recent position paper Papamarkou et. al (2024) discusses that BNNs do not automatically guarantee optimal calibration of uncertainties out of the box. Also, following an illustration in Figure 5  from Deshpande et. al (2023) published recently in TMLR, even samplers that approximate the same underlying true posterior having everything else unchanged will not always lead (under practical fixed sample sizes, i.e. outside the Bernstein–von Mises regime) to better calibration of predictions than more poor approximations of the posterior of the same model. In that sense, we think that while checking predictions and calibration is an important sanity check, we should be a bit cautious in claiming based on empirical evidence that a specific approximate method is improving over another method. That said, checking predictive metrics and calibrations are very important sanity checks. We now have added them (see response to Reviewer 1 for more details). Shortly summarizing, we added ECE calibrations for classification and Pinball losses for regression tasks, where overall, our methods demonstrate robust performance which is definitely not worse than that of the compared approaches (on average one may even claim with caution that slightly better). The simulation of Gaussians 2d example is just an illustration of that the uncertainty reduces as sample size increases in line with Bernstein–von Mises based behavior, which  we would want as the sample size increases. This does not happen with MC dropout. We also illustrate in this example that we can detect out of distribution cases. We are clear that it is not a comparative but rather illustrative example.
> >
> >
> > Comment: The authors did not compare calibration to HMC or GPs
> >
> > Our response:  Regarding GP and BNNs fitted with HMC, we did not manage to add these approaches as treatment in Figure 3 of the manuscript, so that exactly one design change is present from some other approach.This is so, because GP is a different mode, and for BNN trained with HMC we had to reduce the number of neurons and change the activation from relu to smooth tanh. l But we added results based on the HMC fitted BNNs and GP to the Appendix. Below, we discuss what has been done.
> >
> >  a. We added GP baselines for all regression datasets using the out of the box package from Petri et. al (2023). GP “wins” for one dataset (wine quality data), but in general our method is on par with it.
> >
> > b. HMCs for BNNs appeared more tricky in our experience. We ran HMCs on reduced models with the same priors as in BNN and BNN-MNF, but with 10 neurons in the hidden layer and smooth tanh activations. Even for reduced models, we could not reach desired Rhat statistics using the Stan HMC sampler (perceived as a reliable HMC implementation by the Bayesian stats community) from Sharaf et. al (2020). Neither tuning of the parameters of the sampler nor increasing the number of chains did help, hence we went with the default in the package options and single chains per data fold were run for 100000 iterations. The results were unsatisfactory in terms of Rhat, and not surprisingly lead to poor predictive and calibration results in our case. The results on classification were even more poor. They are not reported in the revised paper. Establishing good mixing of standard HMC in multimodal settings can be tricky as shown in Lan et. al (2014), see Figure 1 there. They say “Compared to random walk Metropolis, standard HMC explores the target distribution more efficiently by exploiting its geometric properties. However, it too tends to fail when the target distribution is multimodal since the modes are separated by high energy barriers (low probability regions) ” As discussed in Papamarkou et. al (2024), BNN posteriors are multimodal due to symmetries in the model.  We clearly did not manage to achieve good mixing in our HMC experiments in BNNs settings. Yet we made an effort. The code is in the supplementaries, and we will be happy to rerun with improved mixing if better tuning parameters are suggested. We did not even try to run HMCs on images as according to Neal (2020) for some relevant image data, “The 500 iterations (each scanning through the training cases 200 times) took 11 days, using an RTX A4000 GPU.” We do not have resources corresponding to TPUs and hence would not be able within reasonable time to rerun all 3 of our image datasets on 10 separate seeds each and using multiple HMC chains.

---

> > > ### Author Response · Authors · 2024-03-25
> > > **Response to Reviewer 3 (K8JC). Part 3**
> > >
> > > Regarding miscellaneous points:
> > >
> > >
> > > What is  above eq 1? Why is  called the mean vector?
> > >
> > > -    Apologies for being unclear. This is the vector of parameters of the  probability distribution of the responses. Now it is fixed.
> > >
> > > There's a sum over k  missing in the eq above eq 9
> > >
> > > -    Thanks. We have fixed this
> > >
> > > hard-tanh is not standard terminology; please define it in the text, e.g., as a footnote.
> > >  We added a link to the PyTorch definition in a footnote.
> > >
> > > What do the square brackets mean in the sum in the first equation in section 4?
> > >
> > > -    Thanks for noticing, we fixed the notation here and used () when appropriate, while keeping [] for the operators.
> > >
> > > The brackets used for the expectation and variance in section 4 are inconsistent, sometimes round, sometimes square.
> > >
> > > -    Thanks for noticing, we have now consistently fixed to use [] for expectations along the paper.
> > >
> > > Most figures, especially Figure 3, are blurry and pixelated; please change them to vector graphics.
> > >
> > > -We changed to vector graphics where possible, thanks.
> > >
> > > Switching from BNN to MNF in Table 1 is misleading; an "MNF" is also a BNN. Hence, I suggest the authors change it to MNF-BNN.
> > >
> > > -    Thanks, we changed to BNN-FLOW for parsimony with LBBNN-FLOW that we used.
> > >
> > > What is  in the TPR and FPR definitions?
> > >
> > > -    They were defined in the equations after table 3 in the original submission as standard  TPR, FPR estimated through repeating the variable selection simulations many times on different seeds, but with fixed data. In the revised paper, we removed them as reviewer 1 mentioned those are known and should be omitted.
> > >
> > >
> > > Carbonetto, P., & Stephens, M. (2012). Scalable variational inference for Bayesian variable selection in regression, and its accuracy in genetic association studies.
> > >
> > > T. Papamarkou, M. Skoularidou, K. Palla, L. Aitchison, J. Arbel, D. Dunson, M. Filippone, V. Fortuin, P. Hennig, A. Hubin, et al. Position paper: Bayesian deep learning in the age of large-scale ai. arXiv preprint arXiv:2402.00809, 2024.
> > >
> > > S. Lan, J. Streets, and B. Shahbaba. Wormhole hamiltonian monte carlo. In Proceedings of the AAAI Conference on Artificial Intelligence, volume 28, 2014.
> > >
> > > Varvia, P., Räty, J., & Packalen, P. (2023). mgpr: An R package for multivariate Gaussian process regression. SoftwareX, 24, 101563.
> > >
> > > Deshpande, S., Ghosh, S., Nguyen, T. D., & Broderick, T. (2023). Are you using test log-likelihood correctly?. Transactions on Machine Learning Research.
> > >
> > > Hubin, A., & Storvik, G. (2024). Sparse Bayesian Neural Networks: Bridging Model and Parameter Uncertainty through Scalable Variational Inference. Mathematics, 12(6), 788.
> > >
> > > Sharaf, T., Williams, T., Chehade, A., & Pokhrel, K. (2020). BLNN: An R package for training neural networks using Bayesian inference. SoftwareX, 11, 100432.

---

> > > > ### Comment · Reviewer_K8JC · 2024-03-26
> > > > **Response to the authors**
> > > >
> > > > I thank the authors for their detailed rebuttal and effort to incorporate our feedback into their updated manuscript. They now motivate their work much better, especially in the introduction, and hence, I now believe the paper would be a good fit for TMLR. I have three more significant outstanding issues and a few minor ones before I am ready to recommend acceptance.
> > > >
> > > > ## Major:
> > > >  - I appreciate the authors' conducting experiments with HMC-sampled BNNs. However, they should omit the HMC experiments from the paper due to the compute limitations they mention in their rebuttal. This is because the HMC results in the paper are incomparable to the other results since the authors use a significantly weaker architecture. Consequently, the conclusion drawn from these experiments in the main text, which claims that "[HMC] was underperforming" (bottom of page 9), is invalid and misleading.
> > > >  - Similarly, I appreciate the Gaussian process regression experiments; however, there is no description of the model the authors used, and hence, the numbers they obtained are not interpretable. They should at least include what covariance function they used and whether they performed exact GP regression or opted for some approximation.
> > > >  - Every plot in the paper (Figures 5-10) is still pixelated. Please update them to a vector graphics format.
> > > >
> > > > ## Minor:
> > > >  - "Earlier work (Hubin & Storvik, 2019; Bai et al., 2020; Hubin & Storvik, 2024), have considered similar settings approaches, our main contributions are" - This newly added sentence sounds strange.
> > > >  - I'm unsure what the authors now mean by "fully variational Bayesian model averaging." Do they mean that they are "fully variational" or "fully Bayesian"? I am not sure what "fully variational" would mean, and I would disagree with the claim that their method is "fully Bayesian" since they are doing variational inference and also do not integrate out the hyperparameters.
> > > >  - "Due to a more proper procedure for handling uncertainty, the Bayesian approach does, in many cases, result in more reliable solutions with less overfitting and better uncertainty measures." - I'm not sure what the authors mean by better uncertainty measures; I am not aware of any mainstream frequentist (or other) approach that attempts to quantify parameter uncertainty other than the Bayesian approach.
> > > >  - The starting sentence of section 2 sounds quite strange, as $f$ is called a function and a distribution in the same sentence.
> > > >  - Eq (10) follows directly from applying the chain rule for relative entropies; there is no need to reference another paper.
> > > >  - Please do not redirect to a website for a definition of hard-tanh; define it in the footnote.
> > > >  - In Table 9, the first appearance of  BNN-HMC should just read "BNN," I think.

---

> > > > > ### Author Response · Authors · 2024-03-27
> > > > > **Response from the authors**
> > > > >
> > > > > We gratefully thank the reviewer for the constructive and thorough feedback. We ensure that if the paper is accepted, all the major and minor points raised by you above will be fixed in the camera ready version.

---

### Decision · Action_Editor_WKMM · 2024-04-12

**Recommendation:** Reject

**Comment:**

The algorithm proposed here proposed on a combination of two existing tricks. The reviewers agree that the idea is simple (almost straightforward, to be honest) but clever: [A9N5: "Combining existing ideas obviously feels somewhat incremental"], [WWin: "This is a straight forward idea to try (...) the straightforwardness of the idea I think is a strength"]. [K8JC] shares this opinion.

There are many possible ways to defend such a simple idea: to prove it works with theoretical arguments and/or to provide strong empirical evidence. The authors chose the experimental way. First, the experimental results are good. However, they don't show a clear improvement over existing methods. The results are totally comparable in terms of predictive accuracy, which is expected [WWin: "Obviously, testing something these days on the MNIST family of models doesn't provide a huge amount of evidence of superiority of a method"]. [A9N5] also points out a lack of calibration check on real data, and in the regression setting.

Of course, the benefit expected from the Bayesian approach is uncertainty quantification. But this would be true from many methods (any other Bayesian approach, dropbout etc.) and here again, there is no evidence that the proposed method improves on them. [K8JC: "I am OK with the authors' argument that outright predictive accuracy shouldn't be the primary goal, but my impression is existing BNNs do no better than simpler approaches (dropout, deep ensembles) for predictive uncertainty and still require post-hoc calibration. I have no idea from the paper whether these extensions are helping address this"]? To carricature things, the experiments look like a routine sanity check, but they do not bring any striking evidence of a benefit due to the new algorithm proposed here: [K8JC: "the experiments are standard and do not seem to give any new insight"].

It must be noted that the authors took all these comments very seriously. They provided a detailed reply to all three reviewers together with a deeply revised version of the paper. The new version includes regression examples [A9N5: "The authors did a solid job addressing my concerns"]. This led [A9N5] to (weakly) support publication. The authors also included comparison to standard methods (HMM, GP) for uncertainty quantification. However [K8JC] still believe that this part is not convicing yet (the comparison to HMM seems to be unfair while it's not clear what GP algorithm is used, etc). [K8JC] clearly states he/she might ultimately support publication of the paper, but only after a major revision. Finally, [WWin] recommends to accept the paper.

It is thus a tough call. The paper is clearly boarderline. But I believe that the experimental evidence must be stronger to justify publication. I will thus follow reviewer [K8JC] and reject the paper.

However, in view of the relatively positive opinion of [WWin,A9N5] I strongly encourage the authors to take into account the additional comments/questions of [K8JC] and to resubmit the paper to TMLR.

**Audience:**

Potentially large audience (all the Bayesian ML community, beyond Bayesian deep learning).

**Claims And Evidence:**

The authors propose a new approach to sparsify Bayesian neural networks learnt. This is typically done with spike-and-slab priors (and posteriors when one uses variational approximations). The authors include latent variables that will enfore some weights to be in the same "state" (spike, or slab), and thus to be simultateously set to 0 or not. This latent structure fits nicely with the normalizing flow used in the algorithm. The algorithm is tested on the MNIST dataset and logistic regression with synthetic data. They claim that the results match the state of the art in terms of accuracy, with the benefit of uncertainty quantification due to the Bayesian approach. A problem pointed out by the referee is the lack of quantitative comparison to other existing approaches providing uncertainty quantification.

**Resubmission Of Major Revision:**

The authors may consider submitting a major revision at a later time.